# Collaborative World Models: An Online-Offline Transfer RL Approach

## Abstract

Training offline reinforcement learning (RL) models with visual inputs is challenging due to the coupling of overfitting issue in representation learning and the risk of overestimating true value functions. Recent work has attempted to alleviate the overestimation bias by encouraging conservative behaviors beyond the scope of the offline dataset. This paper, in contrast, tries to build flexible constraints for the offline policies without impeding the exploration of potential advantages. The key idea is to leverage an off-the-shelf RL simulator, with which can be easily interacted in an online manner. In this auxiliary domain, we perform an actor-critic algorithm whose value model is aligned to the target data and thus serves as a "*test bed*" for the offline policies. In this way, the online simulator can be used as the *playground* for the offline agent, allowing for mildly-conservative value estimation. Experimental results demonstrate the remarkable effectiveness of our approach in challenging environments such as DeepMind Control, Meta-World, and RoboDesk. It outperforms existing offline visual RL approaches by substantial margins.

## 1 Introduction

Learning control policies in the physical world through visual observations can be challenging due to high environmental interaction costs and low sample efficiency (Laskin et al., 2020; Schwarzer et al., 2021; Stooke et al., 2021; Xiao et al., 2022; Parisi et al., 2022; Ma et al., 2023). Offline reinforcement learning (RL) has emerged as a promising approach to overcome these challenges. However, applying offline RL in the visual world presents two primary interrelated difficulties. First, offline visual RL may suffer from overfitting issues when extracting hidden states from the limited and partially observable visual data, which may result in trivial solutions of representation learning. Second, similar to its state-space counterpart (Fujimoto et al., 2019; Kumar et al., 2020; Qi et al., 2022; Chen et al., 2023; Zhuang et al., 2023), offline visual RL is susceptible to the challenge of value function overestimation (as observed in existing visual RL methods (Laskin et al., 2020; Hafner et al., 2021)) or underestimation (when excessively constraining values beyond the offline data distribution). Improving offline visual RL remains a relatively under-explored area of research.

Our goal is to address the trade-off between value function overestimation and over-conservatism in offline visual RL by training a policy that is mildly conservative, that is, we should not overly penalize the exploration of actions with potential advantages. To achieve this, as illustrated in Figure 1, we propose to leverage a readily available online simulator for related visual control tasks to train an auxiliary RL agent. Specifically, we present a model-based transfer RL approach called Collaborative World Models (CoWorld). The fundamental idea is to solve offline visual RL as an online-to-offline transfer learning problem. Initially, our approach alleviates the discrepancy between the learned Markov Decision Processes (MDPs) of the source and target domains. By doing so, the source agent effectively serves as an online "*test bed*" for assessing the target policy. It introduces a mild regularization term into the training objective of the target value, which prevents the target agent from overestimating or underestimating the true value function.

Furthermore, harnessing an auxiliary domain provides an additional benefit: it can help alleviate the overfitting issues during representation learning within the offline dataset. With its capability to actively interact with the online environment and accumulate rich information, the source world model, through the alignment of latent state distributions, keeps the target model from trivial solutions when inferring states within the offline domain. Instead, it facilitates the extraction of more generalizable representations from visual observations.

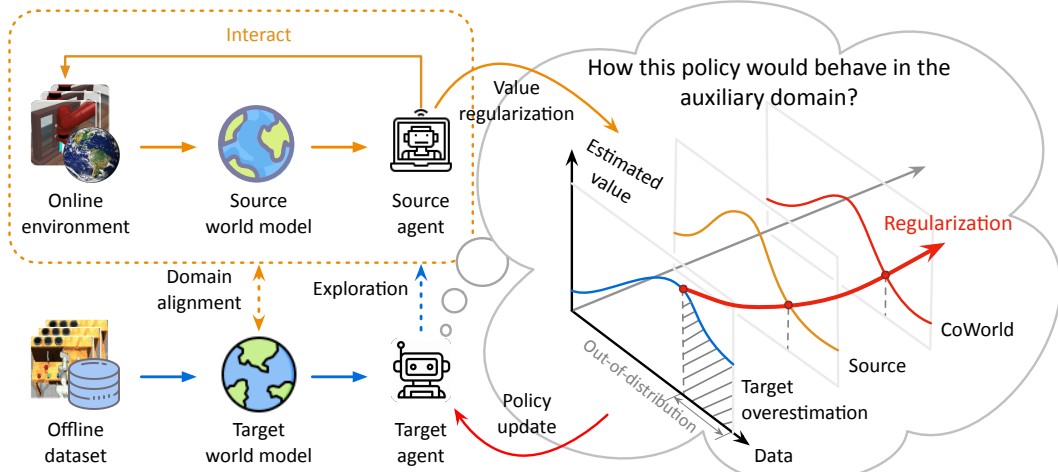

Figure 1: CoWorld solves offline visual RL as an online-to-offline transfer learning problem. The key idea is to leverage a readily accessible online environment as the "*playground*" for the offline agent. The auxiliary world model and the corresponding source agent provide mild constraints for target value estimation, without impeding the exploration of actions with potential advantages.

The key technical contributions of this paper can be summarized as follows:

- First, we propose a novel online-to-offline transfer RL problem, which aims to improve offline visual RL by leveraging an online simulator as the auxiliary domain.

- Second, we present CoWorld, a new model-based RL approach tailored for the online-to-offline setup. CoWorld effectively transfers domain-sharing knowledge by addressing cross-domain discrepancies. Concretely, it aligns the latent spaces across domains and then conducts target-inclined source model tuning and min-max value regularization to improve the offline policy.

Our experiments demonstrate that CoWorld significantly outperforms existing methods in offline visual control tasks in various environments, including DeepMind Control, Meta-World, and RoboDesk. Our approach can effectively mitigate cross-domain discrepancies between the source and the target MDPs, improving offline visual RL by leveraging domain transfer capabilities across different tasks or even across different physical environments. Furthermore, our approach is shown to be easily extended to scenarios with multiple source domains.

## 2 PROBLEM FORMULATION

We consider offline visual RL within the framework of partially observable Markov decision processes (POMDPs) and aim to use a source POMDP denoted by $\langle \mathcal{O}^{(S)}, \mathcal{A}^{(S)}, \mathcal{T}^{(S)}, \mathcal{R}^{(S)}, \gamma \rangle$ to facilitate policy learning in the target POMDP denoted by $\langle \mathcal{O}^{(T)}, \mathcal{A}^{(T)}, \mathcal{T}^{(T)}, \mathcal{R}^{(T)}, \gamma \rangle$. $\mathcal{O}$ denotes the space of visual observations, $\mathcal{A}$ denotes the space of actions, $\mathcal{T}$ denotes the conditional transition probabilities between states, $\mathcal{R}$ is the reward function, and $\gamma \in [0, 1)$ is the discount factor. Our goal is to learn an offline RL agent to maximize the cumulative reward in a fixed dataset $\mathcal{B}^{(T)}$. In our study, we specifically focus on the scenario where a source POMDP is available online, allowing the agent to actively interact with the environment and collect new data.

## 3 METHOD

In this section, we present the technical details of CoWorld, a domain-collaborative, model-based RL algorithm characterized by a pair of world models $\{\mathcal{M}^{(S)}, \mathcal{M}^{(T)}\}$, actor networks $\{\pi^{(S)}, \pi^{(T)}\}$, and critic networks $\{v^{(S)}, v^{(T)}\}$. The fundamental idea driving this algorithm is to solve offline visual RL as an online-to-offline transfer learning problem. Given that all elements within the set $\{\mathcal{O}, \mathcal{A}, \mathcal{T}, \mathcal{R}\}$ may potentially hold distinct distributions across different domains, the entire training process of CoWorld is structured into three iterative stages to mitigate the cross-domain discrepancies:

a) Train the world models while concurrently aligning the latent state space of $\mathcal{M}^{(T)}$ with that of $\mathcal{M}^{(S)}$ (see Section 3.1).

b) Incorporate the target reward information into $\mathcal{M}^{(S)}$, followed by the execution of target-inclined behavior learning of $\pi^{(S)}$ and $v^{(S)}$ in the auxiliary environment (see Section 3.2).

c) Perform model-based behavior learning in the offline domain, where we use the source critic $v^{(S)}$ to provide regularization for $v^{(T)}$ and to assess the rollouts from the policy $\pi^{(T)}$ (see Section 3.3).

### 3.1 DOMAIN-COLLABORATIVE REPRESENTATION LEARNING

The domain-collaborative representation learning process aligns the latent state spaces of the online and offline world models, which in turn contributes to the transferability in value estimation.

**Source model pretraining.** We start with a warm-up phase for the source agent in the auxiliary domain, which is built upon a model-based actor-critic method named DreamerV2 (Hafner et al., 2021). To facilitate subsequent transfer learning, we modify DreamerV2 with an additional state alignment module denoted as $g(\cdot)$. The world model $\mathcal{M}_{\phi'}^{(S)}$ consists of the following components:

$$
\begin{array}{llll}
\text{Recurrent transition:} & h_t = f_{\phi'}^{(S)}(h_{t-1}, z_{t-1}, a_{t-1}) & \text{Image encoding:} & \tilde{z}_t = e_{\phi'}^{(S)}(o_t) \\
\text{Posterior state:} & z_t \sim q_{\phi'}^{(S)}(z_t \mid h_t, \tilde{z}_t) & \text{Prior state:} & \hat{z}_t \sim p_{\phi'}^{(S)}(\hat{z}_t \mid h_t) \\
\text{Reconstruction:} & \hat{o}_t \sim p_{\phi'}^{(S)}(\hat{o}_t \mid h_t, z_t) & \text{Reward prediction:} & \hat{r}_t \sim r_{\phi'}^{(S)}(\hat{r}_t \mid h_t, z_t) \\
\text{Alignment state:} & s_t = g(\tilde{z}_t) & \text{Discount factor:} & \hat{\gamma}_t \sim p_{\phi'}^{(S)}(\hat{\gamma}_t \mid h_t, z_t),
\end{array}
\tag{1}
$$

where $\phi'$ represents the combined parameters of the world model, and $g(\cdot)$ is implemented with the softmax operation. We train $\mathcal{M}_{\phi'}^{(S)}$ on the source replay buffer $\mathcal{B}^{(S)}$ by minimizing

$$
\begin{aligned}
\mathcal{L}(\phi') \doteq \mathbb{E}_{q_{\phi'}^{(S)}} \Big[ \sum_{t=1}^{T} & \underbrace{-\ln p_{\phi'}^{(S)}(o_t \mid h_t, z_t)}_{\text{image reconstruction}} \underbrace{-\ln r_{\phi'}^{(S)}(r_t \mid h_t, z_t)}_{\text{reward prediction}} \underbrace{-\ln p_{\phi'}^{(S)}(\gamma_t \mid h_t, z_t)}_{\text{discount prediction}} \\
& + \underbrace{\text{KL}\Big[ q_{\phi'}^{(S)}(z_t \mid h_t, o_t) \parallel p_{\phi'}^{(S)}(z_t \mid h_t) \Big]}_{\text{KL divergence}} \Big].
\end{aligned}
\tag{2}
$$

We train the actor $\pi_{\psi'}^{(S)}(\hat{a}_t \mid \hat{z}_t)$ and the critic $v_{\xi'}^{(S)}(\hat{z}_t)$ with the respective objectives of maximizing and estimating the expected imagined rewards $\mathbb{E}_{p_{\phi'}, p_{\psi'}}[\sum_{\tau \geq t} \hat{\gamma}_{\tau-t} \hat{r}_\tau]$ generated by $\mathcal{M}_{\phi'}^{(S)}$. Please refer to Appendix A.2 for details of the objectives. We then deploy $\pi_{\psi'}^{(S)}$ to interact with the auxiliary environment and collect new data for further world model training.

**Latent space alignment.** A straightforward transfer learning solution is to train the target agent in the offline dataset upon the checkpoints of the source agent. However, it may suffer from a potential mismatch issue due to the discrepancy in tasks, visual observations, physical dynamics, and action spaces across various domains. In our experiment, this issue becomes more severe when the online data can only be collected within an environment that differs from the one for the offline dataset (*e.g.*, Meta-World → RoboDesk). We tackle this issue by separating the parameters of the source and the target agents while explicitly aligning their latent state spaces. Concretely, we feed the same target domain observations sampled from the offline buffer $\mathcal{B}^{(T)}$ into the two world models and close the distance between $p(s_t^{(S)} \mid o_t^{(T)})$ and $p(s_t^{(T)} \mid o_t^{(T)})$. We optimize $\mathcal{M}_\phi^{(T)}$ by

$$
\begin{aligned}
\mathcal{L}(\phi) \doteq \mathbb{E}_{q_\phi^{(T)}} \Big[ \sum_{t=1}^{T} & \underbrace{-\ln p_\phi^{(T)}(o_t \mid h_t, z_t)}_{\text{image reconstruction}} \underbrace{-\ln r_\phi^{(T)}(r_t \mid h_t, z_t)}_{\text{reward prediction}} \underbrace{-\ln p_\phi^{(T)}(\gamma_t \mid h_t, z_t)}_{\text{discount prediction}} \\
& + \underbrace{\beta_1 \text{KL}\Big[ q_\phi^{(T)}(z_t \mid h_t, o_t) \parallel p_\phi^{(T)}(z_t \mid h_t) \Big]}_{\text{KL divergence}} + \underbrace{\beta_2 \text{KL}\Big[ \text{sg}(g(e_{\phi'}^{(S)}(o_t))) \parallel g(e_\phi^{(T)}(o_t)) \Big]}_{\text{domain alignment loss}} \Big],
\end{aligned}
\tag{3}
$$

where $\text{sg}(\cdot)$ indicates gradient stopping and we use $o_t$ to represent $o_t^{(T)}$ for simplicity. We minimize the Kullback-Leibler (KL) divergence between the latent state distributions produced by $e_\phi^{(T)}(o_t^{(T)})$ and $e_{\phi'}^{(S)}(o_t^{(T)})$, which serves as the foundation for the following learning process. As the source world model can actively interact with the online environment and gather rich information, it keeps the target world model from overfitting the offline visual dataset. $\beta_2$ governs the importance of this loss term. We provide hyperparameter analyses in the appendix.

---

**Algorithm 1:** The training scheme of CoWorld for offline visual RL.

---

1 **Require:** Offline dataset $\mathcal{B}^{(T)}$, imagination horizon $H$, value regularization weight $\alpha$, target-inclined reward factor $k$.

2 **Initialize:** Source and target agent parameters $\{\phi, \phi', \psi, \psi', \xi, \xi'\}$; Replay buffers $\{\mathcal{B}^{(S)}, \mathcal{B}^{(T)}\}$.

3 Pretrain the source agent parametrized by $\{\phi', \psi', \xi'\}$ on the online simulator.

4 **while** *not converged* **do**

5     `// On the offline dataset:`

6     **for** *each step in* $\{1 : K_1\}$ **do**

7         Draw a batch of trajectories from the offline buffer, $\{(o_t, a_t, r_t)\}_{t=1}^T \sim \mathcal{B}^{(T)}$.

8         `// a) Latent space alignment`

9         Train the target world model $\mathcal{M}_\phi^{(T)}$ using Eq. (3) and update $\phi$.

10         `// c) Behavior learning with min-max value regularization`

11         Generate imagined trajectories $\{(z_i, a_i)\}_{i=t}^{t+H}$ with $\pi_\psi^{(T)}$ and $\mathcal{M}_\phi^{(T)}$.

12         Train the target critic $v_\xi^{(T)}$ regularized by $v_{\xi'}^{(S)}$ over $\{(z_i, a_i)\}_{i=t}^{t+H}$, following Eq. (6).

13         Train the target actor $\pi_\psi^{(T)}$ over $\{(z_i, a_i)\}_{i=t}^{t+H}$, and update $\xi$ and $\psi$.

14     **end**

15     `// On the online simulator:`

16     **for** *each step in* $\{1 : K_2\}$ **do**

17         Draw a batch of trajectories from the source buffer, $\{(o_t, a_t, r_t)\}_{t=1}^T \sim \mathcal{B}^{(S)}$.

18         `// b) Target-inclined source model tuning`

19         Compute the target-inclined source rewards $\{r_t^{(S)}\}_{t=1}^T$ with mixed data, following Eq. (4).

20         Compute the loss for the source reward predictor $\hat{r}_{\phi'}^{(S)}$, following Eq. (5).

21         Train the source world model $\mathcal{M}_{\phi'}^{(S)}$ by combining Eq. (2) and Eq. (5), and update $\phi'$.

22         Train the source actor $\pi_{\psi'}^{(S)}$ and critic $v_{\xi'}^{(S)}$ over the imagined $\{(z_i, a_i)\}_{i=t}^{t+H}$, and update $\psi'$ and $\xi'$.

23         Use $\mathcal{M}_{\phi'}^{(S)}$ and $\pi_{\psi'}^{(S)}$ to interact with the source environment, and append $\mathcal{B}^{(S)}$.

24     **end**

25 **end**

---

## 3.2 TARGET-INCLINED SOURCE MODEL TUNING

To enable the source critic to value the target domain policy, it is essential to provide it with prior knowledge of the offline task. In other words, it is necessary to encourage the source agent to explore trajectories similar to those in the target domain. To achieve this, we train the source reward predictor $r_{\phi'}^{(S)}(\cdot)$ using mixed data from both of the replay buffers $\mathcal{B}^{(S)}$ and $\mathcal{B}^{(T)}$ (**Lines 19-21** in Alg. 1). Through the behavior learning on source domain imaginations (**Line 22** in Alg. 1), the target-inclined reward predictor equips the auxiliary RL agent with the ability to assess the imagined states produced by the target model (as we will discuss in Section 3.3).

Specifically, given a target domain trajectory sampled from $\mathcal{B}^{(T)}$, we use the source domain world model parametrized by $\phi'$ to extract the recurrent state $h_t^{(T)}$ and the posterior state $z_t^{(T)}$. In this way, we can relabel the source domain reward by

$$r_t^{(S)} = \begin{cases} (1-k) \cdot r_{\phi'}^{(S)}(h_t^{(T)}, z_t^{(T)}) + k \cdot r_t^{(T)} & \text{if} \quad (o_t^{(T)}, r_t^{(T)}) \sim \mathcal{B}^{(T)} \\ r_t^{(S)} & \text{if} \quad (o_t^{(S)}, r_t^{(S)}) \sim \mathcal{B}^{(S)} \end{cases}, \qquad (4)$$

where $r_{\phi'}^{(S)}(\cdot)$ is the source domain reward predictor. $k$ is the target-inclined reward factor, which acts as a balance between the true target reward $r_t^{(T)}$ and the output of $r_{\phi'}^{(S)}(\cdot)$ when provided with $o_t^{(T)}$. We train $r_{\phi'}^{(S)}(\cdot)$ by minimizing a maximum likelihood estimation (MLE) loss:

$$\mathcal{L}_r = \mathbb{E}_{\mathcal{B}^{(S)}} \Big[ \sum_{t=1}^T - \ln r_{\phi'}^{(S)}(h_t^{(S)}, z_t^{(S)}) \Big] + \mathbb{E}_{\mathcal{B}^{(T)}} \Big[ \sum_{t=1}^T - \ln r_{\phi'}^{(S)}(h_t^{(T)}, z_t^{(T)}) \Big], \qquad (5)$$

which quantifies the negative log-likelihood of observing the relabelled source reward $r_t^{(S)}$. During this stage, the entire source world model $\mathcal{M}_{\phi'}^{(S)}$ is trained by merging Eq. (5) into Eq. (2).

## 3.3 MIN-MAX TARGET VALUE REGULARIZATION

During the process of target domain behavior learning, we employ the auxiliary source critic to estimate the value function of the imagined states. This enhances the learned behaviors of the target actor model, as outlined in **Lines 11-13** of Alg. 1.

To mitigate the risk of value overestimation or excessive conservatism, we introduce a regularization term to the critic loss. This term aims to minimize the maximum value among the estimates provided by both the source and target critic models:

$$\mathcal{L}(\xi) \doteq \mathbb{E}_{p_\phi, p_\psi} \Big[ \sum_{t=1}^{H-1} \underbrace{\frac{1}{2} \Big( v_\xi^{(T)}(\hat{z}_t) - \mathrm{sg}\big(V_t^{(T)}\big) \Big)^2}_{\text{value regression}} + \underbrace{\alpha \max \Big( v_\xi^{(T)}(\hat{z}_t), \ \mathrm{sg}\big(v_{\xi'}^{(S)}(\hat{z}_t)\big) \Big)}_{\text{regularization}} \Big]. \quad (6)$$

The loss function of the target critic model consists of two parts: (1) fitting cumulative value estimates and (2) *regularizing the overestimated values for out-of-distribution data in a mildly conservative way*. Specifically, in line with DreamerV2 (Hafner et al., 2021), we train $v_\xi^{(T)}(\hat{z}_t)$ to regress $V_t^{(T)}$, whose specific formulation can be located in Appendix A.3. Intuitively, it incorporates a weighted average of reward information over an $n$-step future horizon. The hyperparameter $\alpha$ corresponds to the value regularization weight. We provide a sensitivity analysis on the value of $\alpha$ in Appendix G. We stop the gradient of the source critic to keep it from being influenced by the regularization term.

The *min-max* operation is designed to strike a balance between mitigating value overestimation and avoiding over-conservatism. It achieves this by not directly penalizing the value function of state-action pairs that lie outside the dataset but instead focusing on instances where the values exceed the scope of value estimation provided by the target-inclined source critic:

- When the target value estimate becomes excessively high, that is, $v_\xi^{(T)}(\hat{z}_t) \geq v_{\xi'}^{(S)}(\hat{z}_t)$, the second term in Eq. (6) will tend to reduce this value down to $v_{\xi'}^{(S)}(\hat{z}_t)$. This encourages the target critic to be more conservative and avoid overestimation of the true value function.

- Conversely, when the source critic yields larger values, the min-max regularization term does not contribute to the training of the target critic. This strategy promotes exploration within the target domain rollouts without introducing excessive value conservatism.

## 4 EXPERIMENTS

In this section, we present (i) quantitative comparisons with existing visual RL algorithms; (ii) discussions on the principles of source domain selection; (iii) ablation studies of each proposed training stage; and (iv) further analyses of the value over-/underestimation problems.

### 4.1 EXPERIMENTAL SETUPS

**Datasets.** As shown in Figure 2, we evaluate CoWorld across three RL environments, *i.e.*, Meta-World (Yu et al., 2019), RoboDesk (Kannan et al., 2021), and DeepMind Control Suite (DMC) (Tassa et al., 2018), including both cross-task and cross-environment setups. Our data collection strategy aligns with that in D4RL (Fu et al., 2020), where we build offline datasets of *medium-replay* quality using a DreamerV2 agent (Hafner et al., 2021). The datasets comprise all the samples in the replay buffer collected during the training process until the policy attains medium-level performance, defined

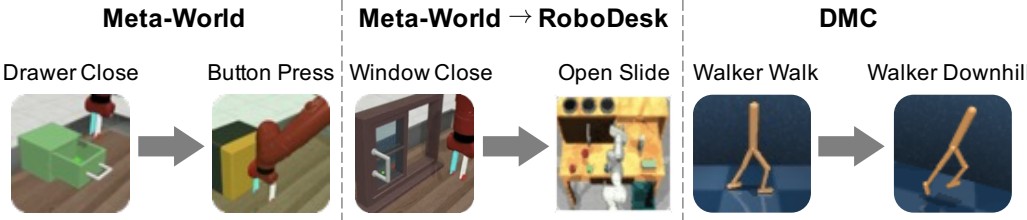

Figure 2: Demonstrations of *cross-task* and *cross-environment* domain transfer. Particularly for the cross-environment experiments, it is important to note that a significant data distribution shift exists in various aspects, including visual observations, action spaces, physical dynamics, and reward functions, which introduces more pronounced out-of-distribution (OOD) challenges for transfer RL.

Table 1: Mean episode returns and standard deviations of 10 episodes over 3 seeds on Meta-World.

| MODEL | BP→ DC* | DC→ BP | BP→ HP | HP→ BT | WC→ DC | BT→ WC | AVG. |
|---|---|---|---|---|---|---|---|
| OFFLINE DV2 | 2143±579 | 3142±533 | 278±128 | 3002±346 | 3899±679 | 3921±752 | 2730 |
| DRQ+BC | 567±19 | 587±68 | 1203±234 | 642±99 | 134±64 | 623±85 | 626 |
| CQL | 1984±13 | 867±330 | 988±39 | 462±67 | 577±121 | 683±268 | 927 |
| CURL | 1972±11 | 51±17 | 986±47 | 189±10 | 366±52 | 281±73 | 641 |
| LOMPO | 2883±183 | 446±458 | 2230±223 | 1961±287 | 2756±331 | 2983±569 | 1712 |
| FINETUNE | 3500±414 | 3564±417 | 3702±451 | 3499±713 | 4273±1327 | 4148±971 | 3781 |
| FINETUNE+EWC | 2479±1006 | 47±10 | 314±393 | 232±118 | 825±1665 | 451±55 | 736 |
| COWORLD | **3967±312** | 3623±543 | **4570±677** | **3921±212** | **4845±7** | **4521±367** | **4241** |
| MULTI-SOURCE | 3546±634 | **3677±476** | 4460±783 | 3626±275 | 4841±15 | 4507±59 | 4110 |

as achieving $1/3$ of the maximum score that the DreamerV2 agent can achieve. CoWorld also achieves competitive results on *medium-expert* dataset. Please refer to Appendix F for detailed information.

**Compared methods.** We compare CoWorld with both model-based and model-free RL approaches, including *Offline DV2* (Lu et al., 2023), *DrQ+BC* (Lu et al., 2023), *CQL* (Lu et al., 2023), *CURL* (Laskin et al., 2020), and *LOMPO* (Rafailov et al., 2021). In addition, we introduce the *Finetune* method, which involves taking a DreamerV2 (Hafner et al., 2021) model pretrained in the online source domain and subsequently finetuning it in the offline target dataset. Furthermore, *Finetune* can be integrated with the continual learning method, Elastic Weight Consolidation (EWC) (Kirkpatrick et al., 2017), to regularize the model for preserving source domain knowledge, *i.e.*, *Finetune+EWC*. Please refer to Appendix B for more details about the baseline methods.

**Multi-source experiments.** CoWorld can easily be extended to scenarios with multiple source domains by adaptively selecting a useful task as the auxiliary simulator. This extension can be easily achieved by measuring the distance of the latent states between the target domain and each individual source domain. Please refer to Appendix E for the technical details of the *adaptive source domain selection* method. We perform multi-source experiments in two scenarios: (1) among multiple tasks within the Meta-World environment, and (2) using tasks from the Meta-World benchmark as source domains and tasks from the RoboDesk environment as the target domain.

## 4.2 META-WORLD

Meta-World is an open-source simulated benchmark designed for solving a wide range of robot manipulation tasks. We select 6 tasks to serve as either the offline dataset or potential candidates for the online auxiliary domain. These tasks include: *Window Close* (**WC**), *Button Press* (**BP**), *Drawer Close* (**DC**), *Door Close* (**DC***), *Button Topdown* (**BT**), and *Handle Press* (**HP**).

**Qualitative comparisons.** In Table 1, we present the results of CoWorld and other compared models in the Meta-World offline datasets. Notably, CoWorld achieves the best performance in all 6 tasks. Notably, it outperforms *Offline DV2*, a method also built upon DreamerV2 and tailored for offline visual RL, by a substantial margin. Among the baseline methods, *Finetune* achieves the second-best results by leveraging transferred knowledge from the auxiliary source domain. However, we observe that its performance experiences a notable decline in scenarios involving significant data distribution shifts between the source and the target domains in visual observation, physical dynamics, reward definition, or even the action space of the robots, such as in the following cross-environment experiments from Meta-World to RoboDesk (see Figure 4). Another important baseline model is *Finetune+EWC*, which focuses on mitigating the catastrophic forgetting of the knowledge obtained in source domain pretraining. However, without additional model designs for domain adaptation, retaining the source domain knowledge may ultimately impact the performance in the target domain.

**Random source domain.** Figure 3(a) presents the Transfer Matrix of CoWorld among the 6 tasks of Meta-World. Values larger than 1 indicate positive domain transfer effects. Notably, there are challenging domain transfer cases with weakly related source and target tasks, while CoWorld outperforms *Offline DV2* in the majority of cases (26 out of 30).

**Multi-source CoWorld.** As shown in Table 1, the multi-source CoWorld achieves comparable results to the models trained with manually designated online simulators. In Figure 3(a), multi-source CoWorld achieves positive improvements over *Offline DV2* in all cases, approaching the best results of models using each individual source task as the auxiliary domain. In Figure 3(b), multi-source CoWorld consistently outperforms the *Finetune* baseline, even when the single-source CoWorld faces

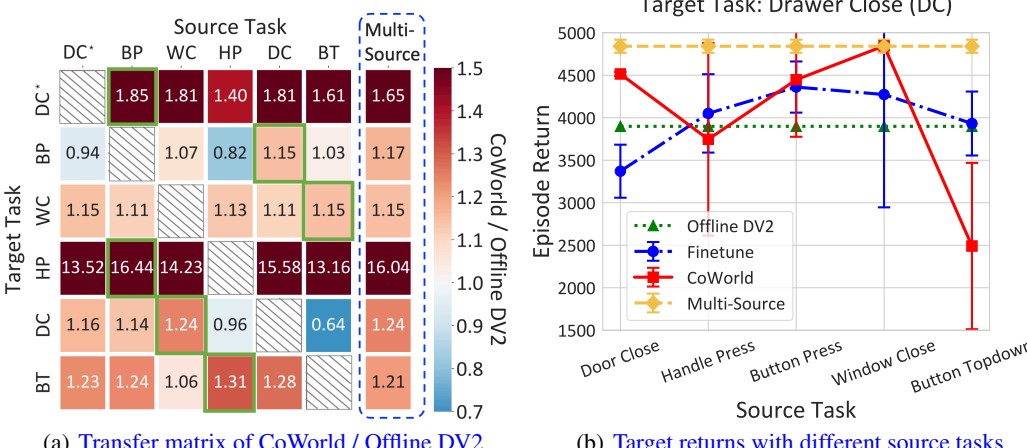

(a) Transfer matrix of CoWorld / Offline DV2        (b) Target returns with different source tasks

Figure 3: (a) The value in each grid cell signifies the ratios of returns achieved by CoWorld compared to those achieved by the *Offline DV2*. Cells highlighted with a green box represent the best-source tasks for transfer. (b) Multi-source CoWorld consistently achieves the best results. The Y-axis represents the episode return for the *Drawer Close* task.

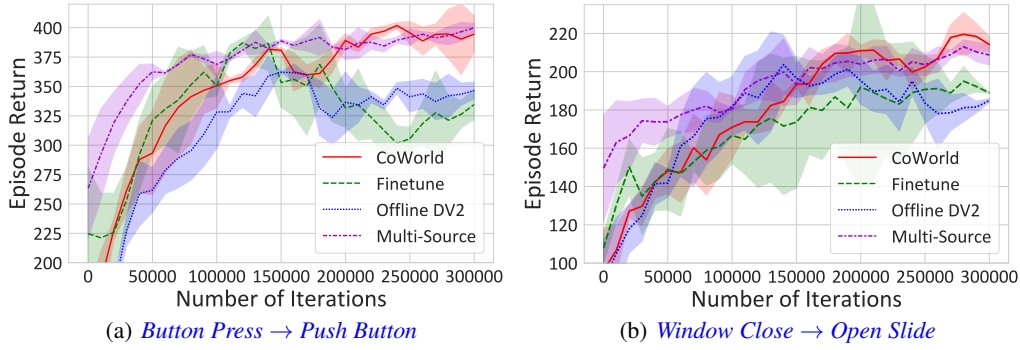

(a) *Button Press → Push Button*        (b) *Window Close → Open Slide*

Figure 4: Quantitative results on the online Meta-World → offline RoboDesk dataset.

challenges with specific undesirable source domains. These results demonstrate our approach's ability to operate without strict assumptions about domain similarity and its ability to automatically identify a useful online simulator from a set of both related and less related source domains.

## 4.3 META-WORLD TO ROBODESK

To explore cross-environment domain transfer, we employ the *Push Green Button* and *Open Slide* tasks from the RoboDesk environment to construct the offline datasets, with Meta-World serving as the source domain. These tasks require handling randomly positioned objects with high-dimensional image inputs. We use *Button Press* as the source task for *Push Green Button* and *Window Close* as the source task for *Open Slide*. For the multi-source experiments, we jointly use three tasks from Meta-World (*i.e.*, *Button Press*, *Window Close*, and *Button Topdown*) as the source domains, and use tasks from RoboDesk (*i.e.*, *Push Green Button* and *Open Slide*) as individual target domains.

Figure 4 presents quantitative results of CoWorld on the RoboDesk dataset, where our approach outperforms *Offline DV2* and *Finetune* by large margins. Unlike the aforementioned results obtained in transfer learning scenarios involving tasks within the same environment, directly fine-tuning the pretrained world model on the target datasets doesn't lead to significant improvements in the final performance. These results can be attributed to the inconsistency in the visual information, physical dynamics, and action spaces (including differences in the dimensions of action signals) between the two environments. In contrast, CoWorld addresses these challenges by leveraging a separate source agent with a domain-specific world model and explicitly aligning the target state representations with those of the source domain, which consequently avoids overfitting. This approach, along with the target-inclined source model tuning method, enables CoWorld to bridge the substantial gap between the two environments and ultimately achieve superior performance. We also showcase the performance of multi-source CoWorld, which achieves comparable results to the *best-source* model that exclusively uses our manually designated auxiliary simulator as the source domain.

Table 2: Mean rewards and standard deviations of 10 episodes in offline DMC over 3 seeds. CoWorld outperforms *Offline DV2* by a remarkable margin of 169.6% and outperforms *DrQ+BC* by 37.5%.

| MODEL | WW → WD | WW → WU | WW → WN | CR → CD | CR → CU | CR → CN | AVG. |
|---|---|---|---|---|---|---|---|
| OFFLINE DV2 | 435±22 | 139±4 | 214±4 | 243±7 | 3±1 | 51±4 | 181 |
| DRQ+BC | 291± 10 | 299±15 | 318±40 | 663±15 | 202±12 | 132±33 | 355 |
| CQL | 46±19 | 64±32 | 29±2 | 2±1 | 52 ±57 | 111±157 | 51 |
| CURL | 43±5 | 21±3 | 23±3 | 26±7 | 4±2 | 11±4 | 21 |
| LOMPO | 462 ± 87 | 260±21 | **460±9** | 395±52 | 46±19 | 120±4 | 291 |
| FINETUNE | 379±23 | 354±29 | 407±37 | 702±41 | 208±22 | 454±82 | 417 |
| COWORLD | **629 ± 9** | **407±141** | 426±32 | **745±28** | **225 ±20** | **493 ±10** | **488** |

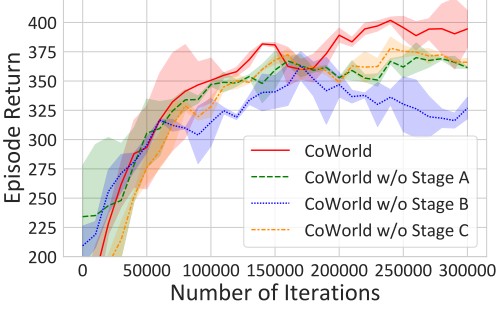
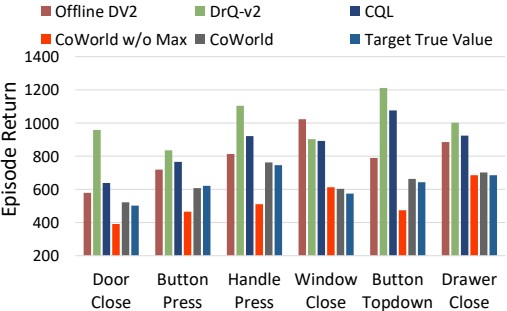

(a) Ablation studies on *Push Green Button* (RoboDesk)  (b) Value estimations with different models

Figure 5: **(a)** Ablation studies of latent space alignment (green), target-inclined source model tuning (purple), and min-max target value regularization (orange). **(b)** The true values and the values estimated by different critics on Meta-World. The values are rescaled for better visualization.

## 4.4 DEEPMIND CONTROL SUITE

DMC is a widely explored benchmark for continuous control. We use the *Walker* and *Cheetah* as the base agents and make modifications to the environment to create a set of 8 distinct tasks, *i.e.*, *Walker Walk* (**WW**), *Walker Downhill* (**WD**), *Walker Uphill* (**WU**), *Walker Nofoot* (**WN**), *Cheetah Run* (**CR**), *Cheetah Downhill* (**CD**), *Cheetah Uphill* (**CU**), *Cheetah Nopaw* (**CN**). Particularly, *Walker Nofoot* is a task in which we cannot control the right foot of the *Walker* agent. *Cheetah Nopaw* is a task in which we cannot control the front paw of the *Cheetah* agent.

We apply the proposed multi-source domain selection method to build the domain transfer settings shown in Table 2. It is worth noting that CoWorld outperforms the other compared models in 5 out of 6 DMC offline datasets, and achieves the second-best performance in the remaining task. On average, it outperforms *Offline DV2* by 169.6% and outperforms *DrQ+BC* by 37.5%. Corresponding qualitative comparisons can be found in the supplementary materials.

## 4.5 FURTHER ANALYSES

**Ablation studies.** We conduct a series of ablation studies to validate the effectiveness of latent space alignment (**Stage A**), target-inclined source model tuning (**Stage B**), and min-max target value regularization (**Stage C**). We show corresponding results on the offline *Push Green Button* dataset from RoboDesk in Figure 5(a). It is evident that the performance experiences a significant decline when we abandon each individual training stage in CoWorld.

**Value estimation analyses.** We evaluate the values estimated by the critic network of CoWorld on the offline Meta-World datasets when the training process is finished. In Figure 5(b), we compute the cumulative value predictions over a span of 500 steps and compare these results with the true value and the values estimated by the other models. The *true value* is determined by calculating the discounted sum of the actual rewards obtained by the actor in the same 500-steps period. We observe that existing approaches, including *Offline DV2* and *CQL*, often overestimate the value functions in the offline visual RL setup. The baseline model, labeled as "*CoWorld w/o Max*", is a variant of CoWorld that incorporates a brute-force constraint on the critic loss. It directly penalizes the target values and reformulates Eq. (6) as $\sum_{t=1}^{H-1} \frac{1}{2}(v_\xi^{(T)}(\hat{z}_t) - \text{sg}(V_t^{(T)}))^2 + \alpha v_\xi^{(T)}(\hat{z}_t)$, excluding both the max operator and the regularization term introduced by $v_{\xi'}^{(S)}(\hat{z}_t)$. As observed, this model tends to

underestimate the true value function, which can potentially result in overly conservative policies as a consequence. In contrast, the values estimated by CoWorld are notably more accurate and more akin to the true values. This validates the effectiveness of the proposed min-max target value regularization approach in mitigating the overestimation or over-conservatism problems during offline training.

## 5  RELATED WORK

**Offline RL.**  Offline RL utilizes pre-collected offline data to optimize policies. It faces a fundamental limitation in that offline RL agents lack the ability to interact with the online environment, leading to OOD issues, as discussed by Levine et al. (2020). Model-free offline RL methods, such as BCQ (Fujimoto et al., 2019), CQL (Kumar et al., 2020), and LAPO (Chen et al., 2022), mainly suggest taking actions that were previously present in the offline dataset or learning conservative value estimations. Model-based methods, such as MOPO (Yu et al., 2020), COMBO (Yu et al., 2021), and RAMBO (Rigter et al., 2022), use an ensemble dynamics model to predict the transition process and also incorporate conservative techniques to mitigate the OOD problem. While many existing offline RL approaches primarily target low-dimensional state problems, recent approaches have introduced offline visual RL methods that enable learning policies directly from high-dimensional images (Mandlekar et al., 2019; Dasari et al., 2019; Levine et al., 2020; Agarwal et al., 2020; Rafailov et al., 2021; Yu et al., 2022; Seo et al., 2022; Zang et al., 2023).

**Visual RL.**  While most previous literature in RL focuses on tasks with compact state representations, the ability to learn directly from rich observation spaces like images is critical for real-world applications. Model-based approaches have demonstrated their effectiveness in visual RL (Hafner et al., 2019; 2020; 2021; Seo et al., 2022; Pan et al., 2022; Hafner et al., 2022; Mazzaglia et al., 2023; Micheli et al., 2023; Zhang et al., 2023; Ying et al., 2023). Furthermore, Rafailov et al. (2021); Cho et al. (2022); Lu et al. (2023) have introduced specific techniques to address the challenges associated with offline visual RL. Rafailov et al. (2021) proposed to handle high-dimensional observations with latent dynamics models and uncertainty quantification. Cho et al. (2022) proposed synthesizing the raw observation data to append the training buffer, aiming to mitigate the issue of overfitting. In a related study, Lu et al. (2023) modified DreamerV2 (Hafner et al., 2021) to establish competitive offline visual baselines for continuous control. In contrast to the aforementioned approaches, CoWorld presents a novel idea that borrows an auxiliary online simulator to improve offline visual RL.

**Transfer RL.**  In real-life scenarios, it is essential for agents to utilize the knowledge learned in past tasks to facilitate learning in unseen tasks, which is known as transfer RL (Zhu et al., 2020; Sekar et al., 2020; Zhang et al., 2020; Sun et al., 2021; Zhang et al., 2021; Eysenbach et al., 2021; Yang & Nachum, 2021; Sun et al., 2022; Ghosh et al., 2023; Kumar et al., 2023; Rafailov et al., 2023; Liu et al., 2023; Nakamoto et al., 2023). In the context intersected with visual RL, CtrlFormer (Mu et al., 2022) learns a transferable state representation via a sample-efficient vision Transformer. APV (Seo et al., 2022) executes action-free world model pretraining on source-domain videos and finetunes the model on downstream tasks. Choreographer (Mazzaglia et al., 2023) builds a model-based agent that exploits its world model to learn and adapt skills in imaginations, the learned skills are adapted to new domains using a meta-controller. VIP (Ma et al., 2023) presents a self-supervised, goal-conditioned value-function objective. This objective operates independently of actions, enabling the use of unlabeled video data during pretraining. Unlike the above literature, our work presents an online-to-offline framework that ultimately facilitates offline learning performance.

## 6  CONCLUSIONS AND DISCUSSIONS

In this paper, we proposed a transfer RL method named CoWorld, which mainly tackles the difficulty in representation learning and value estimation in offline visual RL. The key idea is to exploit a readily accessible online environment to train an auxiliary RL agent. To address domain discrepancies and to improve the offline policy, specific technical contributions include 1) latent space alignment, 2) target-inclined source model tuning, and 3) min-max value regularization. CoWorld demonstrates competitive results across three RL benchmarks and is shown effective in multi-source scenarios.

Nonetheless, it remains crucial to acknowledge that an unsolved problem in CoWorld is the increased computational complexity associated with the training phase in the auxiliary source domain. Improving the training efficiency of CoWorld is a valuable avenue for future research.

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

# A    MODEL DETAILS

## A.1    FRAMEWORK OF COWORLD

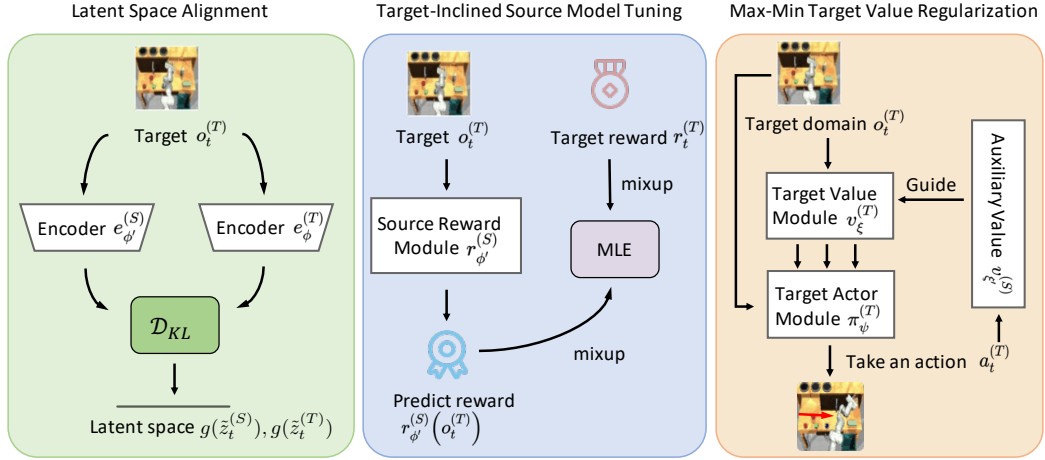

Figure 6: CoWorld uses an auxiliary online environment to build a policy "*test bed*" that is aware of offline domain information. This, in turn, can guide the visual RL agent in the offline domain to learn a mildly-conservative policy, striking a balance between value overestimation and over-conservatism.

## A.2    WORLD MODEL LEARNING

We adopt the framework of the world model used in Hafner et al. (2021). The image encoder is a Convolutional Neural Network (CNN). The image predictor is a transposed CNN and the transition, reward, and discount factor predictors are Multi-Layer Perceptrons (MLPs). The discount factor predictor serves as an estimate of the probability that an episode will conclude while learning behavior based on model predictions. The encoder and decoder take $64 \times 64$ images as inputs.

The loss function of the target world model (*i.e.*, Eq. (3)) is jointly minimized with respect to the $\phi'$ that contains all parameters of the target world model. The image predictor, reward predictor, discount predictor, and transition predictor are trained to maximize the log-likelihood of their individual targets through the distributions they produce.

## A.3    BEHAVIOR LEARNING

For behavior learning of CoWorld, we use the actor-critic learning architecture of DreamerV2 (Hafner et al., 2021). The $\lambda$-target $V_t^{(T)}$ in Eq. (6) is defined as follows:

$$V_t^{(T)} \doteq \hat{r}_t^{(T)} + \hat{\gamma}_t \begin{cases} (1 - \lambda)v_\xi^{(T)}(\hat{z}_{t+1}) + \lambda V_{t+1}^{(T)} & \text{if } t < H \\ v_\xi^{(T)}(\hat{z}_H) & \text{if } t = H \end{cases}, \tag{7}$$

where $\lambda$ is set to 0.95 for considering more on long horizon targets. The actor and critic are both MLPs with ELU (Clevert et al., 2015) activations while the world model is fixed. The target actor and critic are trained with guidance from the source critic, and regress the $\lambda$-return with a squared loss. The sg in Eq. (6) is a stop gradient function, we stop the gradients around the source critic.

# B    COMPARED METHODS

We compare CoWorld with several widely used model-based and model-free offline methods.

- **Offline DV2** (Lu et al., 2023): A model-based RL method that modifies DreamerV2 (Hafner et al., 2021) to offline setting, and adds a reward penalty corresponding to the mean disagreement of the dynamics ensemble.

- **DrQ+BC** (Lu et al., 2023): It modifies the policy loss term in DrQ-v2 (Yarats et al., 2021) to match the loss given in Fujimoto & Gu (2021).
- **CQL** (Lu et al., 2023): It is a framework for offline RL that learns a Q-function that guarantees a lower bound for the expected policy value compared to the actual policy value. We add the CQL regularizers to the Q-function update of DrQ-v2 (Kumar et al., 2020).
- **CURL** (Laskin et al., 2020): It is a model-free RL approach that extracts high-level features from raw pixels utilizing contrastive learning.
- **LOMPO** (Rafailov et al., 2021): An offline model-based RL algorithm that handles high-dimensional observations with latent dynamics models and uncertainty quantification.
- **Finetune**: It pretrains a DreamerV2 agent in the online source domain and subsequently finetunes the pretrained agent in the offline target domain. Notably, Meta-World → RoboDesk tasks' action space is inconsistent, and we can't finetune directly. Instead, we use the maximum action space of both environments as the shared policy output dimension. For Meta-World and Meta-World → RoboDesk transfer tasks, pretrain the agent 160k steps, and finetune it 300k steps. For DMC transfer tasks, pretrain the agent 600k steps, and finetune it 600k steps.
- **Finetune+EWC**: It modifies *finetune* method with EWC to regularize the model for retaining knowledge from the online source domain. The steps of pretraining and finetuning are consistent with *Finetune*.

## C IMPLEMENTATION DETAILS.

**Meta-World.** For Meta-World environment, we adopt robotic control tasks with complex visual dynamics. For instance, the *Door Close* task requires the agent to close a door with a revolving joint while randomizing the door positions, and the *Handle Press* task involves pressing a handle down while randomizing the handle positions. To evaluate the performance of CoWorld on these tasks, we compare it with several baselines in six visual RL transfer tasks.

**RoboDesk.** In our study, we select Meta-World as the source domain and RoboDesk as the target domain. Notably, there exists a significant domain gap between these two environments. The visual observations, physical dynamics and action spaces of two environments are different. First, Meta-World adopts a side viewpoint, while RoboDesk utilizes a top viewpoint. Further, the action space of Meta-World is 4 dimensional, while RoboDesk comprises a 5-dimensional action space. Considering these differences, the Meta-World → RoboDesk presents a challenging task for transfer learning.

**DMC.** Source agents are trained with standard DMC environments, target agents are trained in modified DMC environments. In this modified environment, *Walker Uphill* and *Cheetah Uphill* represent the task in which the plane is a $15°$ uphill slope. *Walker Downhill* and *Cheetah Downhill* represents the task in which the plane is a $15°$ downhill slope. We evaluate our model with baselines in six tasks with different source domains and target domains.

## D ASSUMPTIONS OF THE SIMILARITY BETWEEN THE SOURCE AND TARGET DOMAINS

This assumption can be softened by our proposed approaches, namely, 1) latent space alignment and 2) target-inclined source model tuning. Through these approaches, we aim to mitigate domain discrepancies between distinct source and target MDPs.

Empirical evidence supporting our methods is presented in Section 4.3, where our proposed approach demonstrates robust performance in the "Meta-World to RoboDesk" transfer RL setup. The similarities and discrepancies of these two environments are presented in Table 3.

Our experiments reveal that the CoWorld method exhibits a notable tolerance to inter-domain differences in visual observation, physical dynamics, reward definition, or even the action space of the robots. This characteristic makes it more convenient to choose an auxiliary simulator based on the type of robot. For example:

Table 3: The similarities and discrepancies of Meta-World and Robodesk environments.

|  | Source: Meta-World | Target: RoboDesk | Similarity / Difference |
|---|---|---|---|
| Task | Window Close | Open Slide | Related manipulation tasks |
| Dynamics | Simulated Sawyer robot arm | Simulated Franka Emika Panda robot arm | Different |
| Action space | Box([-1. -1. -1. -1.], [1. 1. 1. 1.], (4,), float64) | Box(-1, 1, (5,), float32) | Different |
| Reward scale | [0, 1] | [0, 10] | Different |
| Observation | Right-view images | Top-view images | Different view points |

- When the target domain involves a robotic arm (*e.g.*, RoboDesk), an existing robotic arm simulation environment (*e.g.*, Meta-World as used in our paper) can be leveraged as the source domain.
- In scenarios with legged robots, environments like DeepMind Control with Humanoid tasks can serve as suitable auxiliary simulators.
- For target domains related to autonomous driving vehicles, simulation environments like CARLA can be selected.

## E  DETAILS OF MULTI-SOURCE COWORLD

The key idea of adaptive domain selection of multi-source CoWorld is to allocate a set of one-hot weights $\omega_t^{i=1:N}$ to candidate source domains by calculating their KL divergence in the latent state space to the target domain, where $i \in [1, N]$ is the index of each source domain. The adaptive domain selection procedure includes the following steps:

1. **World models pretraining.** We pretrain a world model for each source domain individually.
2. **Domain distance measurement.** At each training step in the target domain, we measure the KL divergence between the latent states of the target domain, produced by $e_\phi^{(T)}(o_t^{(T)})$, and corresponding states in each source domain, produced by $e_{\phi_i'}^{(S)}(o_t^{(T)})$. Here, $e_\phi^{(T)}$ is the encoder of the target world model and $e_{\phi_i'}^{(S)}$ is the encoder of the world model for source domain $i$.
3. **Auxiliary domain identification.** We dynamically identify the closest source domain with the smallest KL divergence. We set $\omega_t^{i=1:N}$ as a one-hot vector, where $\omega_t^i = 1$ indicates the selected auxiliary domain.
4. **Rest of training.** With the one-hot weights, we continue the rest of the proposed online-to-offline training approach. For example, during representation learning, we adaptively align the target state space to the selected online simulator by rewriting the domain alignment loss term in Eq. (3) as

$$\mathcal{L}_{\text{M-S}} = \beta_2 \sum_{i=1}^N \omega_i \text{KL} \left[ \text{sg}(g(e_{\phi_i'}^{(S)}(o_t))) \, \| \, g(e_\phi^{(T)}(o_t)) \right]. \qquad (8)$$

To evaluate the effectiveness of the multi-source adaptive selection algorithm, we conducted experiments on Meta-World and RoboDesk Benchmark. For each target task, two source tasks are utilized, including the CoWorld best-performing task and the CoWorld worst-performing task. Additionally, the sub-optimal source task is added for some target tasks.

The performance of multi-source CoWorld is reported in Table 1. Experiment results demonstrate that this method can adaptively select the best source task for most multi-source problems to ensure adequate knowledge transfer.

We observe that CoWorld can flexibly adapt to the transfer learning scenarios with multiple source domains, achieving comparable results to the model that exclusively uses our manually designated auxiliary simulator as the source domain (best source). This study significantly improves the applicability of CoWorld in broader scenarios.

# F ADDITIONAL QUANTITATIVE AND QUALITATIVE RESULTS

**Visualizations on policy evaluation.** We evaluate the trained agent of different models on the Meta-World and DMC tasks and select the first 45 frames for comparison. Figure 7 and Figure 8 present showcases of performing the learned policies of different models on DMC and Meta-World respectively.

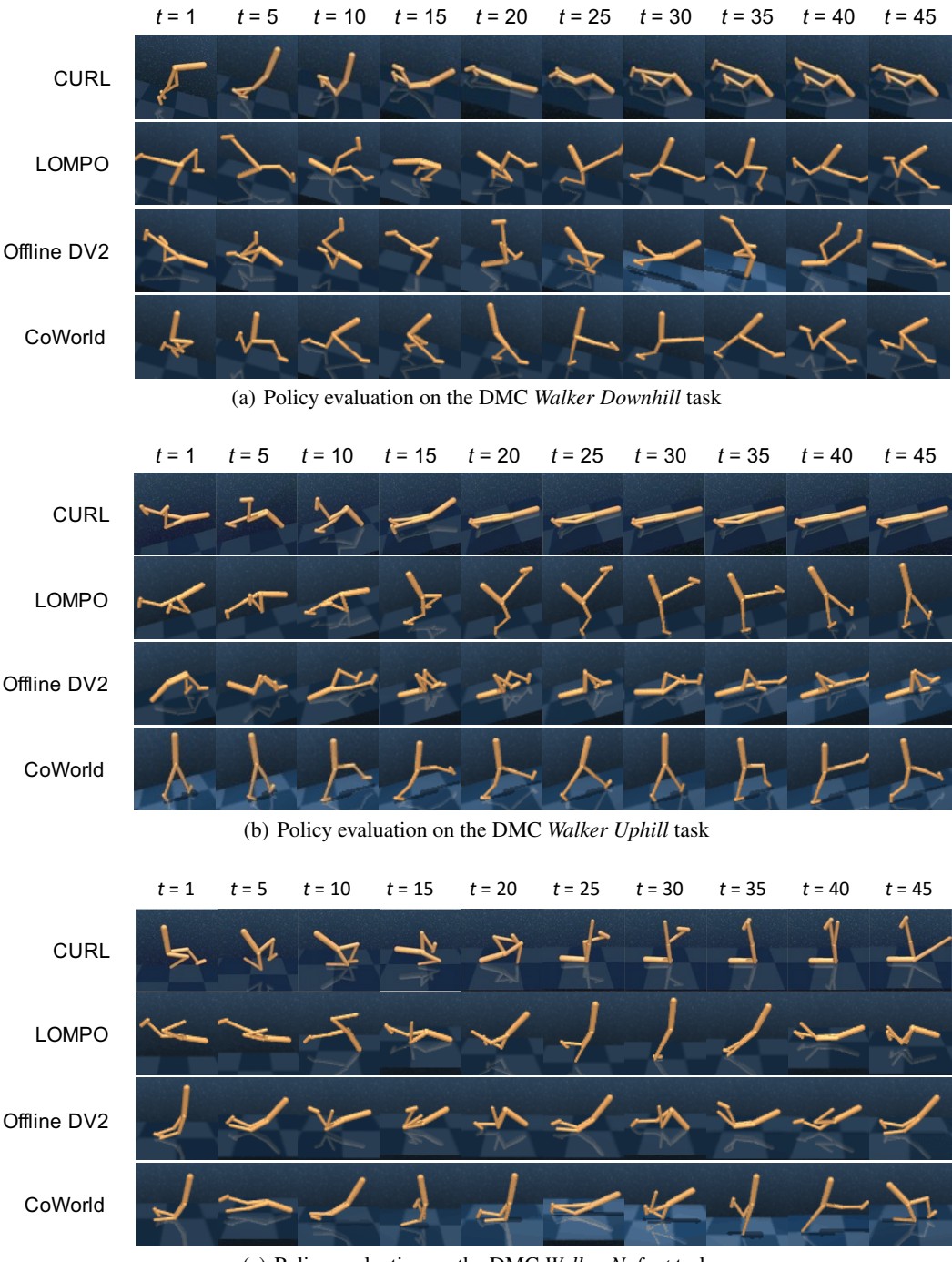

(a) Policy evaluation on the DMC *Walker Downhill* task

(b) Policy evaluation on the DMC *Walker Uphill* task

(c) Policy evaluation on the DMC *Walker Nofoot* task

Figure 7: Additional qualitative results of policy evaluation on the DMC tasks.

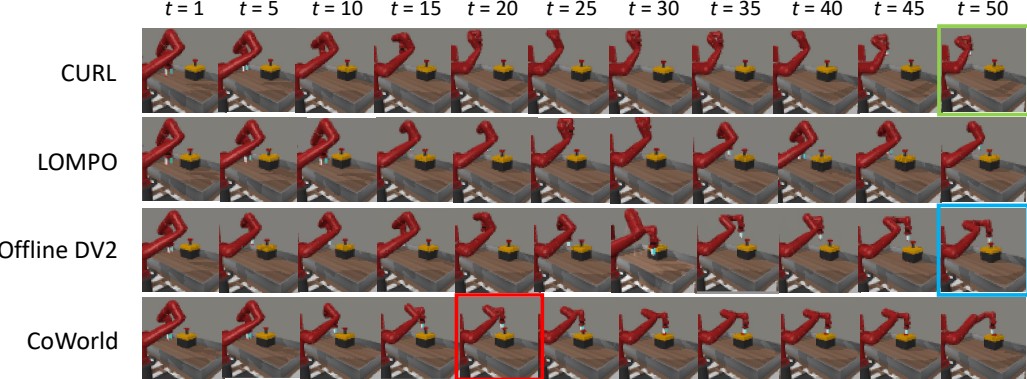

Figure 8: Policy evaluation on the Meta-World *Button Topdown* task. The performance of model-free method *CURL* is poor and it cannot complete the task (green box). CoWorld achieves better performance and completes the task in fewer steps (red box) than *Offline DV2* (blue box).

Table 4: Performance on DMC *medium-expert* dataset. We report the mean rewards and standard deviations of 10 episodes over 3 seeds.

| MODEL | WW → WD | WW → WU | WW → WN | CR → CD | CR → CU | CR → CN | AVG. |
|---|---|---|---|---|---|---|---|
| OFFLINE DV2 | $450 \pm 24$ | $141 \pm 1$ | $214 \pm 8$ | $248 \pm 9$ | $3 \pm 0$ | $48 \pm 3$ | 184 |
| DRQ+BC | $808 \pm 47$ | $762 \pm 61$ | $808 \pm 45$ | $862 \pm 13$ | $454 \pm 12$ | $730 \pm 17$ | 737 |
| LOMPO | $548 \pm 245$ | $449 \pm 117$ | $688 \pm 97$ | $174 \pm 29$ | $19 \pm 10$ | $113 \pm 35$ | 332 |
| FINETUNE | $784 \pm 46$ | $671 \pm 65$ | $851 \pm 91$ | $858 \pm 9$ | $428 \pm 49$ | $\underline{833 \pm 7}$ | $\underline{738}$ |
| COWORLD | $\mathbf{848 \pm 9}$ | $\mathbf{774 \pm 29}$ | $\mathbf{919 \pm 7}$ | $\mathbf{871 \pm 13}$ | $\mathbf{475 \pm 16}$ | $\mathbf{844 \pm 1}$ | **789** |

**Quantitative results on DMC *meidum-expert* dataset.** Similar to the data collection strategy of *medium-replay* dataset, we build offline datasets of *medium-expert* quality using a DreamerV2 agent. The *medium-expert* dataset comprises all the samples in the replay buffer during the training process until the policy attains expert-level of performance, defined as achieving the maximum score that the DreamerV2 agent can achieve. As shown in Table 4, CoWorld outperforms other baselines on the DMC *medium-expert* dataset in most tasks.

**Quantitative results on Meta-World.** Figure 9(a) compares the performance of different models on Meta-World. *Finetune* demonstrates better performance in the initial training phase, thanks to its direct access to the source environment. Instead, CoWorld introduces auxiliary source value guidance to assist the training of the target agent. In the final phase of training, the source value guidance is more effective, and then CoWorld outperforms *Finetune*. Furthermore, Figure 9(b) presents the ablation studies of CoWorld conducted on Meta-World, highlighting the effectiveness and necessity of each training stage.

**Effect of latent space alignment.** We feed the same observations into the source and target encoder of CoWorld and then use the t-distributed stochastic neighbor embedding (t-SNE) method to visualize the latent states. As shown in Figure 10, the representation learning alignment bridges the gap between the hidden state distributions of the source encoder and target encoder.

## G  HYPERPARAMETERS

The hyperparameters of CoWorld are shown in Table 5. We conducted experiments on Meta-World DC → BP task to explore the sensitivity of CoWorld hyperparameters (see Figure 11). We find that when the domain KL loss scale $\beta_2$ is too small, the latent space alignment between the source and target encoders will be inadequate, and it will impair the transfer learning process. Conversely, if the value $\beta_2$ is too large, the target encoder will become excessively inclined towards the source encoder, leading to the performance decline.

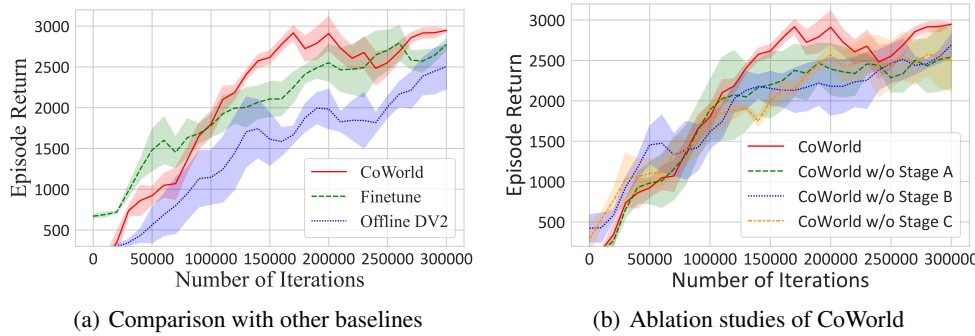

(a) Comparison with other baselines

(b) Ablation studies of CoWorld

Figure 9: **(a)** Comparison with various approaches on the Meta-World *Button Press* task. **(b)** Ablation studies on the Meta-World *Button Press* task that can show the effect of latent space alignment (green), target-inclined source model tuning (purple), and min-max target value regularization (orange).

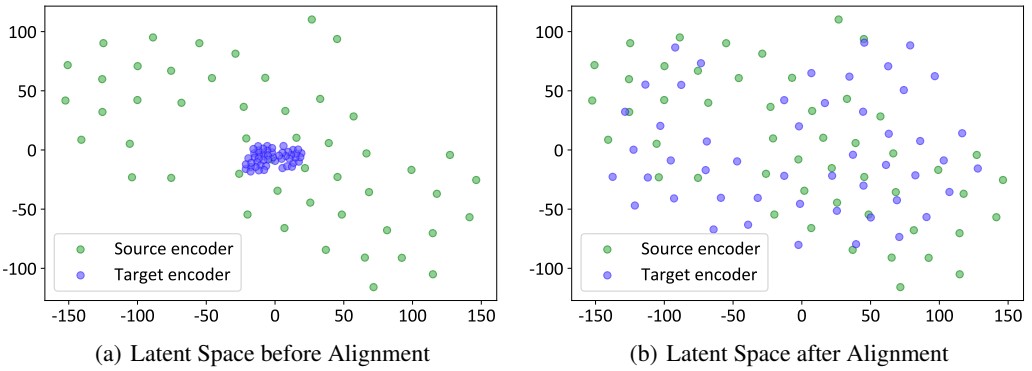

(a) Latent Space before Alignment

(b) Latent Space after Alignment

Figure 10: Visualization of the latent space alignment on Meta-World *Button Press* task by the t-SNE method. **(a)** Latent space of CoWorld before alignment. **(b)** Latent space of CoWorld after alignment.

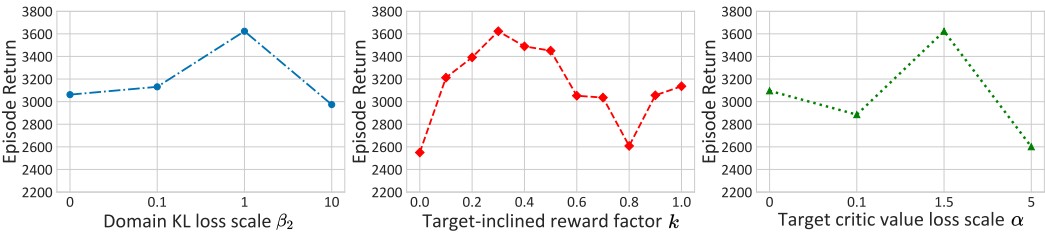

Figure 11: Sensitivity analysis with the hyperparameters of CoWorld on Meta-World DC $\rightarrow$ BP task.

Table 5: Hyperparameters of CoWorld.

| Name | Notation | Value | |
|---|---|---|---|
| Co-training | | Meta-World/RoboDesk | DMC |
| Domain KL loss scale | $\beta_2$ | 1 | 1.5 |
| Target-inclined reward factor | $k$ | 0.3 | 0.9 |
| Target critic value loss scale | $\alpha$ | 1.5 | 1 |
| Source domain update iterations | $K_1$ | $2 \cdot 10^4$ | $2 \cdot 10^4$ |
| Target domain update iterations | $K_2$ | $5 \cdot 10^4$ | $2 \cdot 10^4$ |
| World Model | | | |
| Dataset size | — | $2 \cdot 10^6$ | |
| Batch size | $B$ | 50 | |
| Sequence length | $L$ | 50 | |
| KL loss scale | $\beta_1$ | 1 | |
| Discrete latent dimensions | — | 32 | |
| Discrete latent classes | — | 32 | |
| RSSM number of units | — | 600 | |
| World model learning rate | — | $2 \cdot 10^{-4}$ | |
| Behavior Learning | | | |
| Imagination horizon | $H$ | 15 | |
| Discount | $\gamma$ | 0.995 | |
| $\lambda$-target | $\lambda$ | 0.95 | |
| Actor learning rate | — | $4 \cdot 10^{-5}$ | |
| Critic learning rate | — | $1 \cdot 10^{-4}$ | |

