# OpenReview forum: "Collaborative World Models: An Online-Offline Transfer RL Approach"
_ICLR.cc/2024/Conference — Submitted to ICLR 2024_

### Official Review · Reviewer_77pB · 2023-10-29

**Soundness:** 3 good
**Presentation:** 3 good
**Contribution:** 2 fair
**Rating:** 5
**Confidence:** 4

**Summary:**

This paper studied two challenges of training offline reinforcement learning models with visual inputs: 1) overfitting in representation learning and 2) overestimation issue. The basic idea is to harness a readily available RL simulator such that the offline agent can interact with it while the offline RL can be treated as an online-to-offline transfer learning problem. The interaction with the auxiliary RL agent allows the offline agent to assess the target policy by introducing a  regularization term in the training. Meanwhile, the online interaction is also designed to facilitate the learning of more generalized representations from the visual input. Experimental results presented in the paper demonstrate the effectiveness of this approach, comparing with the models such as CQL and offline DV2 in six robot manipulation tasks.

**Strengths:**

+ The paper is well-motivated, and itaims to study two challenges in offline RL.

+ The paper is overall well-written and provides a detailed description of the proposed collaborative world models (CoWorld).

+  Based on the experimental studies presented in the paper, the proposed CoWorld seems to be a promising approach for offline RL by providing better performance compared with other offline RL methods. Especially it shows to outperform offline DV2 by a large margin.

**Weaknesses:**

One main challenge in the proposed approach is how to choose a source RL simulator. Clearly, the key of the online-to-offline transfer lies in the quality of the source agent or the alignment between the source and target tasks. A priori,  how do you set the criteria for choosing the source task and how do you ensure the source task is sufficiently informative for the target tasks? Needless to say, one cannot  choose it arbitrarily. So how do you specify the threshold according to the values in the transfer matrix?  Also, what is the impact of the source task quality on the learning performance.  Many of these details are missing.

 Some details of the proposed method are missing or simply punted to another paper, e.g., the objective functions in Eqn. (2), Eqn. (6). In order to make this paper self-contained, it is necessary to include those details.

 This work is compared with the fine-tune method, where in offline RL, there are many possible ways to fine-tune the model. It is important to specify the details of these methods, e.g., how many steps for fine-tuning for each task?

**Questions:**

1. When computing the transfer matrix, how many steps of online fine-tune and online learning are used, respectively? Meanwhile, how many steps of fine-tune in Table 1 and Figure 4?

2. What is the variance in Figure 3(b)?

3. In both Figure 5(a), would you explain why in iteration around 150k, the CoWorld performance is worse or similar compared with CoWorld w/o State A/B/C?

4. What is the computational complexity of the training phase of the auxiliary agent?

5. In Figure 5(b), the CQL (Kumar et al.) method seems to overestimate the value function for all the tasks, why is this the case considering that CQL is a conservative way for offline RL?

---

> ### Author Response · Authors · 2023-11-23
> **Responses to Reviewer 77pB (Part 1)**
>
> Thank you for your valuable comments. We hope our responses below can help address your concerns on this paper.
>
> > Q1: How to choose a source RL simulator？ How do you set the criteria for choosing the source task and how do you ensure the source task is sufficiently informative for the target tasks?
>
> We understand the reviewer's concern regarding the impact of the source task quality on the learning performance. We would like to address this question from two perspectives:
>
> (1) CoWorld's performance with a random source domain:
>
> In the revised paper, we have updated **Figure 3(a)** to include the Transfer Matrix among the 6 tasks on the Meta-World benchmark. Within all domain transfer cases, there are instances of pairs with less related source and target tasks. Despite these challenging scenarios, our observations indicate that CoWorld outperforms Offline-DV2 in the majority of cases (26 out of 30).
>
> (2) Results with the **REVISED** "source domain selection" method:
>
> Admittedly, our initial proposal of the "source domain selection" method is not practical in real-world scenarios, as $R_\text{online}$ should not be available during training in the offline RL setup. To address this issue, we have thoroughly revised this method and introduced a new adaptive domain selection approach for scenarios with multiple source domains available.
>
> For detailed technical insights and corresponding results, please refer to our **General Response (Q2)**. In summary, the "adaptive source domain selection" method yields results comparable to models trained with manually designated online simulators. This highlights our approach's ability to automatically identify a useful online simulator from a set of both related and less related source domains, thereby expanding its applicability in practical scenarios.
>
> We have included the aforementioned results in the revised paper.
>
> > Q2: Some details of the proposed method are missing or simply punted to another paper.
>
> Thank you for your suggestion. We have incorporated additional details about the proposed method for world model learning and behavior learning in Appendix A. This includes further explanations regarding the insight behind the objective functions and a more in-depth description of the network architecture.
>
> > Q3: How many steps for fine-tuning for each task?
>
> Below, we present the finetuning steps of the "Finetune" baseline model compared with the training steps of CoWorld in the target dataset. Please refer to Appendix B for more details of the "Finetune" method.
>
> | Method   | Meta-World | DMC  | Meta-World $\rightarrow$ RoboDesk |
> | -------- |:----------:|:----:|:---------------------:|
> | Finetune |    300k    | 600k |    300k    |
> | CoWorld  |    300k    | 600k |   300k   |
>
> > Q4: What is the variance in Figure 3(b)?
>
> We trained the compared models with three random seeds and presented the mean results along with standard deviations. The experimental results in Figure 3(b) have been updated in the revised paper.

---

> ### Author Response · Authors · 2023-11-23
> **Responses to Reviewer 77pB (Part 2)**
>
> > Q5: In Figure 5(a), why CoWorld performance is worse or similar compared with CoWorld w/o State A/B/C?
>
> At approximately 150k iterations, the convergence of the world model and policy in CoWorld is not complete, resulting in unstable performance. We observed a significant decrease in reward variance and evaluation returns around this point, suggesting a potential overfitting process within the reward predictor. However, by continuing the training of CoWorld, we successfully mitigated this overfitting issue. Additionally, around 170k iterations in CoWorld without Stage B, a similar decrease was noted, although without a subsequent increase in reward variance. This discrepancy might have contributed to the final result decline in CoWorld without Stage B.
>
> > Q6: What is the computational complexity of the training phase of the auxiliary agent?
>
> We have provided the comparisons of the training time and the inference time of the compared models in the above **General Response (Q4)**. Please note that our approach yields comparable inference time to the baseline models such as LOMPO.
>
> > Q7: In Figure 5(b), the CQL (Kumar et al.) method seems to overestimate the value function for all the tasks, why is this the case considering that CQL is a conservative way for offline RL?
>
> As there is no official code for CQL with visual inputs, we adopted the DrQ-v2+CQL implementation from the work of V-D4RL (Lu et al., 2023). Below, we compare the value estimated by DrQ-v2+CQL and its base DrQ-v2 model:
>
> | Task | DrQ-v2 Value | DrQ-v2+CQL Value |
> | --- | :---: | :---: |
> | Door Close | 959 | 639 |
> | Button Press | 836 | 766 |
> | Handle Press | 1104 | 921 |
> | Window Close | 903 | 892 |
> | Button Topdown | 1211 | 1076 |
> | Drawer Close | 1001 | 924 |
> | Avg. | 1002 | 870 |
>
> We have used the above results to update Figure 5(b). In comparison to the naive DrQ-v2, using the CQL regularization term can partly mitigate the issue of overestimated values. However, there are still overestimates due to the complexity of offline visual RL problems.
>
> Reference: Lu et al. "Challenges and opportunities in offline reinforcement learning from visual observations", TMLR, 2023.

---

### Official Review · Reviewer_wybs · 2023-10-31

**Soundness:** 2 fair
**Presentation:** 2 fair
**Contribution:** 2 fair
**Rating:** 3
**Confidence:** 2

**Summary:**

This paper utilizes an online RL simulator to facilitate offline RL. The main idea is to learn aligned world models for both source and target domains, and then use the source model and critic to regularize the target critic. This method is supposed to address the trade-off between over-estimation of values and over-conservatism in offline RL. Experiments on cross-task and cross-environment show that the algorithm can transfer knowledge from the source task to the target one.

**Strengths:**

The idea of online-to-offline transfer learning is interesting and sounds novel to me. The exprimental results verify the effectiveness of the proposed method. It is also promising to see the method work for large discrepancy between the source task and the target task.

**Weaknesses:**

- It is not clear how the source domain should be selected in practice. Although the paper provides a selection metric based on the ratio between the finetune model and the online model, it is not practical because we do not have access to the test-time domain (if you have them, why do you do offline learning?) Here the $R_{Online}$ should not be available during training time. It makes the experiment results less convincing, as the source domain selection is actually **leaking information from test to train**.
- The learning process involves alternation between online and offline agents until convergence, which can be expensive and unstable in practice. Can you provide some evidence for the time complexity of the method compared to baselines?
- From the formulation and the algorithm, it is not clear how the source and the target domains are related, which makes it hard to justify the feasibility of the method.

**Questions:**

- For latent state alignment, it is confusing to me why you can feed the same target domain observations into the two world models and close the distance of latent states. Based on the formulation provided, the source domain and the target domain may have different observation spaces $O^{(S)}$ and $O^{(T)}$. Why does it make sense to align $p(s_t^{(S)}|o_t^{(T)})$ and $p(s_t^{(T)}|o_t^{(T)})$? Does it mean you assume the similarity between $s_t^{(S)}$ and $s_t^{(T)}$? The relation between two domains are not clear to me (e.g. how similar are $O^{(S)}$ and $O^{(T)}$). The same issue exists across multiple equations, where the regularization is to align the t-th step of the source and the target, without explaining why they CAN be aligned.
- Can you evaluate the performance of the algorithm without the designed "source domain selection" (for the reason I specified in weakness)? What if the source domain does not match the requirement? Will it hurt the performance by a lot?

---

> ### Author Response · Authors · 2023-11-23
> **Responses to Reviewer wybs**
>
> We appreciate your great efforts in reviewing our paper and hope that the following responses can address most of your concerns.
>
> > Q1: It is not clear how the source domain should be selected in practice, as the source domain selection is actually leaking information from test to train.
>
> Yes, indeed, our initial proposal of the "source domain selection" method is not practical in real-world scenarios. We greatly appreciate that the reviewer pointed it out. To address this problem, we have thoroughly modified the domain selection method, which can extend CoWorld to scenarios with multiple source domains available. This is achieved by measuring the distance of the latent states between the **OFFLINE** target dataset and each source domain provided by different world models. We have included more technical details and corresponding results in **General Response (Q2)**.
>
> In summary, the "adaptive source domain selection" method yields results comparable to models trained with manually designated online simulators. This highlights our approach's ability to automatically identify a useful online simulator from a set of both related and less related source domains, thereby expanding its applicability in practical scenarios.
>
> We have included the aforementioned results in the revised paper.
>
> > Q2: Can you evaluate the performance of the algorithm without the designed "source domain selection"? What if the source domain does not match the requirement? Will it hurt the performance by a lot?
>
> (1) CoWorld's performance with a random source domain:
>
> In the revised paper, we have updated **Figure 3(a)** to include the Transfer Matrix among the 6 tasks on the Meta-World benchmark. Within all domain transfer cases, there are instances of pairs with less related source and target tasks. Despite these challenging scenarios, our observations indicate that CoWorld outperforms Offline-DV2 in the majority of cases (26 out of 30).
>
> (2) Results with the **REVISED** "source domain selection" method:
>
> As discussed above, we have revised the source domain selection method. The updated method can easily extend CoWorld to scenarios with both related and less related source domains by automatically identifying a useful domain as the auxiliary simulator. We observe that:
> - In **Table 1** in the revised manuscipt, the multi-source CoWorld achieves comparable results to the models trained with manually designated online simulators.
> - In **Figure 3(a)**, the multi-source CoWorld achieves positive improvements over Offline-DV2 in all cases, approaching the best results of models using each individual source task as the auxiliary domain.
> - In **Figure 3(b)**, the multi-source CoWorld consistently outperforms the Finetune baseline, even when the single-source CoWorld faces challenges with specific undesirable source domains.
>
> These results demonstrate our approach's ability to operate without strict assumptions about domain similarity.
>
> > Q3: Time complexity of the method compared to baselines.
>
> Please refer to the comparisons of the training/inference time complexity in our **General Response (Q4)**.
>
> > Q4: From the formulation and the algorithm, it is not clear how the source and the target domains are related.
>
> We propose the following two techniques to establish a relationship between the source and target MDPs, mitigating cross-domain discrepancies:
> - **Latent Space Alignment:** To mitigate the cross-domain discrepancies in visual observations. This is achieved by closing the distance of latent states produced by the world models with the same visual inputs.
> - **Target-Inclined Source Model Tuning:** To mitigate the cross-domain discrepancies in reward functions. This is achieved by incorporating the target reward information into the learning process of the source reward predictor. As we assume the latent states in different domains have been successfully aligned, the objective in this stage is to enable the source reward predictor to estimate the target returns based on the aligned state inputs.
>
> > Q5: For latent state alignment, it is confusing to me why you can feed the same target domain observations into the two world models and close the distance of latent states.
>
> In "model transfer" setups, as suggested by Knowledge Flow (Liu et al., 2019) and Transferrable Memory Unit (Yao et al., 2020), it is a common practice to pretrain a teacher model in the source domain, and subsequently transfer the knowledge from the teacher model to the student model in the target domain by providing the teacher model with target domain data. These existing approaches typically assume the presence of distribution shifts between $O^{(S)}$ and $O^{(T)}$.
>
> References:
> - Liu et al. "Knowledge flow: Improve upon your teachers", ICLR, 2019.
> - Yao et al. "Unsupervised transfer learning for spatiotemporal predictive networks", ICML, 2020.

---

### Official Review · Reviewer_MD4J · 2023-10-31

**Soundness:** 2 fair
**Presentation:** 3 good
**Contribution:** 2 fair
**Rating:** 3
**Confidence:** 2

**Summary:**

This paper introduces a new problem statement in which we have access to an online simulator that does not match the distribution of target offline data. The presented CoWorld approach iterates between online training in a source environment and offline training in the target environment and regularizes target world models with critics learned in the source. The authors evaluate both cross-task and cross-environment transfer settings using Meta-World, DeepMind Control Suite, and RoboDesk. CoWorld outperforms offline-only training with DreamerV2, DrQ+BC, CQL, CURL, and LOMPO.

**Strengths:**

This paper includes an interesting discussion on the transfer learning problem within RL and presents a realistic and relevant problem statement that is currently understudied. The CoWorld training framework enables the author to study interesting questions about transfer learning for RL, especially around the difficulty in transferring from one environment or task to another (see Figure 3). The paper shows that for Dreamer-V2 the CoWorld approach improves final performance in the target environment.

**Weaknesses:**

It's not clear is CoWorld is a method or a new problem statement. In my opinion the paper presents two things:

1. A new _problem statement_ where we have a simulator where we can do online RL where our objective is to perform well on an offline dataset from a target environment
2. A method for doing well in this new problem statement

Currently in the paper 1 and 2 are both presented as a method. I think it would be more straightforward and fair to separate these contributions.

This is important because only one baseline is fine-tuned from the source online simulator to the offline data. However, based on prior work on off-line to online fine-tuning, I would expect different RL baselines to have different levels of success when provided access to a online training in a source environment. For example, the IQL [Kostrikov et. al., 2021] paper shows that different offline RL algorithms perform differently when fine-tuned online. I think to have proper baselines, all methods should be able to access the online simulator. It is also important to add baselines that focus specifically on the offline to online problem statement (namely, IQL).

- Kostrikov et. al. Offline Reinforcement Learning with Implicit Q-Learning. 2021.

**Questions:**

To confirm, are the DrQ+BC, CQL, CURL, and LOMPO baselines all trained fully offline without access to the online source environment?

---

> ### Author Response · Authors · 2023-11-23
> **Responses to Reviewer MD4J**
>
> Thank you for the insightful comments.
>
> > Q1: It's not clear is CoWorld is a method or a new problem statement. I think it would be more straightforward and fair to separate these contributions.
>
> Following the reviewer's suggestion, we have refined our introduction to elaborate on the paper's contributions in two key aspects:
> * First, we propose a novel online-to-offline transfer RL problem, which aims to improve offline visual RL by leveraging an online simulator as the auxiliary domain.
> * Second, we present CoWorld, a new model-based RL approach tailored for the online-to-offline setup. CoWorld effectively transfers domain-sharing knowledge by addressing cross-domain discrepancies. This is achieved through:
>     * Latent Space Alignment: To mitigate the domain shifts in visual observations;
>     * Target-Inclined Source Model Tuning: To mitigate the domain shifts in rewards;
>     * Min-Max Value Regularization: To enable mildly-conservative value estimation.
>
> > Q2: Only one baseline is fine-tuned from the source online simulator to the offline data.
>
> First, we have compared CoWorld with more baseline models that are pretained with source domain data and then finetuned in the target dataset. Please refer to our **General Response (Q3)** for the results. We found that all of the baseline models suffer from the so-called "negative transfer" effect when incorporating a source pretraining stage. This indicates that a naive transfer learning scheme might degenerate the target performance by introducing unexpected bias.
>
> Furthermore, we implemented another baseline model by combining the Finetuning method with elastic weight consolidation (EWC) (Kirkpatrick et al., 2017). EWC allows the model for preserving source domain knowledge. However, we found that, without additional model designs for target domain adaptation, maintaining knowledge from the source domain could affect the performance in the target domain. Detailed quantitative comparisons across different Meta-World tasks can be found in the **General Response (Q3)** and **Table 1** in the revised paper.
>
> For the suggested method, IQL, we did not find a well-performed off-the-shelf implementation that allows for visual inputs. We tried the official code for the work from Cho et al. (2022), which includes an implementation of image-based IQL, but found that it did not converge well. Besides, we should note that IQL was originally designed for the offline-to-online problem, which is inherently different from our problem setup.
>
> References:
> - Kirkpatrick et al. "Overcoming catastrophic forgetting in neural networks", Proceedings of the national academy of sciences, 2017.
> - Cho et al. "S2P: State-conditioned image synthesis for data augmentation in offline reinforcement learning", NeurIPS, 2022.
>
>
> > Q3: Are the DrQ+BC, CQL, CURL, and LOMPO baselines all trained fully offline?
>
> Yes, they were. However, as discussed above, we have provided new results by pretraining these models in the source domain. Please refer to the quantitative comparisons in our **General Response (Q3)**.

---

### Official Review · Reviewer_mTFC · 2023-11-01

**Soundness:** 3 good
**Presentation:** 3 good
**Contribution:** 2 fair
**Rating:** 6
**Confidence:** 4

**Summary:**

The paper presents Collaborative World Models (CoWorld), a novel approach to offline reinforcement learning (RL) with visual inputs. CoWorld uses an existing online RL simulator as a 'test bed' for offline policies, allowing for moderate value estimation without hindering the exploration of potentially beneficial actions.

CoWorld treats offline visual RL as an online-to-offline transfer learning problem and is designed to address the trade-off between overestimation and over-conservatism in value functions. The approach involves three stages: training world models and aligning latent state spaces, incorporating target reward information into the source model and performing model-based behavior learning in the offline domain.

Experiments demonstrate that CoWorld significantly outperforms existing methods in offline visual control tasks across various environments, including DeepMind Control, Meta-World, and RoboDesk.

**Strengths:**

1. This paper proposes an online-to-offline transfer learning approach to tackle offline RL with visual inputs. To my best knowledge, this setting is relatively new and understudied.

2. The methodology proposed seems to adequately addresses the main challenges in offline RL with visual inputs. Specifically, DOMAIN-COLLABORATIVE REPRESENTATION LEARNING (stage A) alleviates the overfitting issue in representation learning, and stage B \& C attempts to strike a balance between value function overestimation and over-conservatism.

3. The experiments are relatively comprehensive. In particular, the paper demonstrates results on three RL benchmarks: Meta-World, RoboDesk and DeepMind Control Suite. Moreover, both cross-task and cross-environment domain transfer experiments are conducted.

**Weaknesses:**

1. **Unrealistic assumptions**.  The paper tries to tackle offline RL by assuming the existence of an off-the-shelf simulator from a source domain. On one hand, as discussed in part 4.2, the success of this online-to-offline paradigm highly hinges on the similarity between the source and target domains. On the other hand, offline RL research is motivated by the assumption that online data collection can be too costly or dangerous on the target domain. These two conditions seem contradictory since if we cannot query large amount of online data from a target domain due to safety or economic issues, it's unlikely that we can build a high-quality simulator on a very similar source domain.

2. **Marginal or unclear performance improvement of the method**.  The paper compares CoWorld with a number of offline RL methods trained **on the target domain only** with only one exception, which is a simple transfer learning strategy which finetunes the source model on the target domain. I find the performance improvement of CoWorld not convincing enough due to the following considerations:

-  There are many potentially better baseline methods that can leverage data from both domains, which the paper doesn't compare. To name a few: finetuning the model with early stopping/soft update/elastic weight consolidation [1], or finetuning the model and source agent simultaneously, or simply co-training a policy on both domains simultaneously.

- The performance of CoWorld seems to be highly sensitive to hyperparameters such as the Target-inclined reward factor $k$. According to Figure 11, by simply changing $k=0.3$ to $k=0.7$ results in about 600 decline to ~3000 in episode return. In this case, CoWorld is inferior to both Finetune and Offline DV2 on DC $\rightarrow$ BP task according to Table 1. The improvement in Figure 9 also looks very marginal compared to the naive Finetune baseline.

3. **Computational complexity**: As highlighted in section 6, CoWorld achieves a marginal improvement at the cost of increase computational complexity, which is a notable limitation.

Minor issues:

1. The paper seems rushed. What follows requires further clarification:

- In figure 4, what are the source and target domains for (a) and (b)? Why do you conclude that "performance experiences a notable decline in scenarios involving a significant data distribution shift between the source and the target domains, such as in the following cross-environment experiments from Meta-World to RoboDesk"?

[1] Kirkpatrick, James, et al. "Overcoming catastrophic forgetting in neural networks." Proceedings of the national academy of sciences 114.13 (2017): 3521-3526.

**Questions:**

1. Your method relies on the availability of an online RL simulator. Could you elaborate on how dependent the performance of CoWorld is on the quality of this simulator? How would the method perform if the simulator is not a perfect representation of the real-world task?

2. For all experiments, competitive model-free methods such as CQL consistently underperform model-based methods by a large margin. However, we generally expect that model-free methods achieve higher asymptotic performance compared to model-based methods at the cost of sample efficiency. Any explanations?

---

> ### Author Response · Authors · 2023-11-23
> **Responses to Reviewer mTFG (Part 1)**
>
> We greatly appreciate your valuable comments. We hope the responses below can help address your concerns.
>
> > Q1: In Figure 4, what are the source and target domains for (a) and (b)?
>
> Figure 4 provides cross-environment results, where the source domains (a: Press Button, b: Close Window) are from the Meta-World environment, while the target domains (a: Push Button, b: Open Slide) are from another environment named Robodesk.
>
> The data distribution shifts between the source domain and the target domain reside in visual observation, physical dynamics, reward definition, and even the action space of the robots. Let's take the experimental setup in Figure 4(a) for example:
>
> |              | Source: Meta-World                                   | Target: RoboDesk             | Similarity / Difference                                                                     |
> |:------------ |:---------------------------------------------------- |:---------------------------- |:---------------------------------------------------------------------------------- |
> | Task         | Window Close                                        | Open Slide                   | Related manipulation tasks |
> | Dynamics     | Simulated Sawyer robot arm      | Simulated Franka Emika Panda robot arm      | Different                                                                   |
> | Action space | Box([-1. -1. -1. -1.], [1. 1. 1. 1.], (4,), float64) | Box(-1, 1, (5,), np.float32) | Different                                                                   |
> | Reward scale | [0, 1]                                               | [0, 10]                      | Different                                                                   |
> | Observation  | Right-view images                               | Top-view images         | Different viewpoints, scene object, and backgrounds |
>
> > Q2: Could you elaborate on how dependent the performance of CoWorld is on the quality of this simulator? How would the method perform if the simulator is not a perfect representation of the real-world task?
>
> We understand the reviewer's concern regarding the dependencies between the source domain and the target domain. We would like to address this question from two perspectives:
>
> First, it is important to emphasize that the cross-environment experiments (Meta-World to RoboDesk) we clarified in the above response are **INDEED** the scenarios that "the simulator is not a perfect representation of the real-world task". Our experimental results in **Figure 4** reveal that **our approach can effectively mitigate the cross-domain discrepancies** in visual observation, physical dynamics, reward definition, or even the action space of the robots. In methodology, this is achieved by 1) Latent Space Alignment and 2) Target-Inclined Source Model Tuning.
>
> Second, in **Figure 3(a)** in the revised paper, we have updated the Transfer Matrix among the 6 tasks on the Meta-World benchmark. Among all domain transfer cases, there exist pairs of seemingly unrelated source and target tasks. However, we observe that our approach outperforms Offline-DV2 in the majority of scenarios (26/30 cases).
> > Q3: Unrealistic assumptions --- the success of this online-to-offline paradigm highly hinges on the similarity between the source and target domains... It's unlikely that we can build a high-quality simulator on a very similar source domain.
>
> As discussed earlier, CoWorld does not rely on strict assumptions about domain similarity. The results presented in Figure 3 and Figure 4 illustrate CoWorld's effectiveness in mitigating cross-domain discrepancies and utilizing a less related online simulator to improve offline RL.
>
> Furthermore, CoWorld can be easily extended to online-to-offline scenarios with multiple source domains. As detailed in our **General Response (Q2)**, CoWorld achieves comparable results to the models trained with manually designated online simulators. This demonstrates our approach's ability to automatically identify a useful online simulator from a set of both related and less related source domains, thereby expanding its applicability in practical scenarios.

---

> > ### Author Response · Authors · 2023-11-23
> > **Responses to Reviewer mTFG (Part 2)**
> >
> > > Q4: The paper compares CoWorld with a number of offline RL methods trained on the target domain only with only one exception. There are many potentially better baseline methods that can leverage data from both domains.
> >
> > Following the reviewer's suggestion, we additionally compare CoWorld with the following baselines:
> > - **More Finetuning Baselines:** We pretrained the baseline models, including Offline DV2, DrQ+BC, and LOMPO, using the source domain data, and then finetuned the models on the target dataset. We found that all of the baseline models suffer from the so-called "negative transfer" effect when incorporating a source pretraining stage. --- A naive transfer learning scheme might degenerate the target performance by introducing unexpected bias. Please refer to our **General Response (Q3)** for details of the quantitative results.
> > - **Finetune+EWC:** We introduced a baseline model utilizing the continual learning approach, EWC, to enforce regularization for preserving knowledge from the online source domain. However, our observations reveal that, without additional model designs for domain adaptation, maintaining knowledge from the source domain could affect the performance in the target domain.
> > - **One World Model on Both Domains:** We implemented a baseline model named "Multi-Task DV2", which involves training DreamerV2 on both offline and online data with a joint world model and separate actor-critic models. As we can see from the results below, CoWorld consistently performs better.
> > - **One Policy Model on Both Domains:** We also implemented the "OnePolicy" model, which employs separate world models while learning one shared policy across both source and target domains.
> >
> > Here, we summarize the results of the newly added baseline models for the cross-environment experiments (Meta-World to RoboDesk). We present the mean returns and standard deviations that are calculated over 10 episodes with 3 random seeds. It is important to note that all of these models can access the source domain data as CoWorld does, and our approach still achieves the best performance.
> >
> > | Meta-World $\rightarrow$ RoboDesk | Button Press  $\rightarrow$ Push Button | Window Close $\rightarrow$ Open Silde |
> > |:--------------------------------- |:--------------------------------------- |:------------------------------------- |
> > | Multi-Task DV2                    | 342 $\pm$ 29                            | 173 $\pm$   22                        |
> > | Finetune+EWC                      | 259 $\pm$ 14                            | 161 $\pm$   6                         |
> > | OnePolicy                         | 364 $\pm$  51                           | 186 $\pm$  27                         |
> > | CoWorld                           | **428 $\pm$ 42**                            | **202 $\pm$ 19**                          |
> >
> > We have included these comparisons in the revised paper.
> >
> > > Q5: Hyperparameter sensitivity of $k$.
> >
> > We have refined the hyperparameter analysis results in Figure 11, providing a more detailed exploration of $k$. The results below showcase the robustness of CoWorld in the range of $0.1 \le k \le 0.5$, achieving more than 3200 episode returns on average.
> >
> > | $k$ | 0 | 0.1 | 0.2 | 0.3 | 0.4 | 0.5 | 0.6 | 0.7 | 0.8 | 0.9 | 1 |
> > | --- | --- | --- | --- | --- | --- | --- | --- | --- | --- | --- | --- |
> > | CoWorld's Returns | 2550 $\pm$ 493 | **3212 $\pm$ 801** | **3392 $\pm$ 771** | **3623 $\pm$ 543** | **3489 $\pm$ 751** | **3451 $\pm$ 463** | 3052 $\pm$ 1138 | 3036 $\pm$ 698 | 2608 $\pm$ 998 | 3056 $\pm$ 623 | 3136 $\pm$ 867 |
> >
> > We'll provide more analyses with finer intervals on the other two hyperparameters.
> >
> > > Q6: Computational complexity.
> >
> > We have provided the comparisons of the training time and the inference time of the compared models in the above **General Response (Q4)**. Please note that our approach yields comparable inference time to the baseline models such as LOMPO.
> >
> > > Q7: Why do the model-free methods such as CQL consistently underperform model-based methods by a large margin?
> >
> > It is an interesting question, and we will explain this in two aspects:
> > - First, in the visual continuous control tasks on the Meta-World/DMC/RoboDesk benchmarks, the base model, DreamerV2, typically outperforms the model-free methods such as CURL and DrQ-V2 (i.e., our base model for CQL). Such results are illustrated in Figure 7 from the arXiv version of the DrQ-V2 paper ("Mastering Visual Continuous Control: Improved Data-Augmented Reinforcement Learning" by Yarats et al., 2021).
> > - Second, in offline visual RL scenarios, model-free approaches are more likely to be affected by the overfitting problem when learning latent states from limited offline visual observations. This challenge can be partly alleviated by the model-based approaches, as the "imagination" process in the policy learning procedure in DreamerV2 (and our approach as well) can naturally augment the offline data.

---

### Official Review · Reviewer_G38A · 2023-11-08

**Soundness:** 3 good
**Presentation:** 3 good
**Contribution:** 2 fair
**Rating:** 5
**Confidence:** 2

**Summary:**

The authors introduce Collaborative World Models (CoWorld), a method for offline reinforcement learning (RL) that conceptualizes the challenge as an online-to-offline transfer learning problem. CoWorld aims to reduce the representation learning issue of overfitting that emerges from using limited visual data and mitigate the common tendency of value function overestimation in offline RL.

The method utilizes an auxiliary online simulator and learns separate source and target world models and source and target actor-critic agents, each with its own policy and value function. Regularization of the target agent value function is designed to prevent over-estimation without being overly conservative. They tackle the representation overfitting problem by aligning the target and source model's latent spaces. CoWorld iteratively performs a three-stage learning procedure: aligning the latent spaces between source and target world models, target-inclined source model tuning, and behavior learning with min-max value regularization.

**Strengths:**

- *Strategic Integration of Transfer Learning*: The authors utilize an online simulator to serve as a 'test-bed" and gather more data from an analogous task. This parallels multi-task and meta-learning approaches that rely on transfer learning across tasks.

- *Empirical Validation*: Experimental results show that the method works well. They also demonstrate the effectiveness of a simple pretraining with fine-tuning baseline, which suggests that transfer between tasks is crucial.

- *Cohesive Methodological Design*: The components of CoWorld contribute synergistically to the overall performance enhancement, as validated by ablation studies.

**Weaknesses:**

- *Dependence on Auxiliary Simulators:* The CoWorld framework's performance is critically dependent on the availability of an online simulator in a domain that is sufficiently similar to the target domain. This reliance may limit the method's applicability in situations where such simulators are unavailable or where the source and target domains do not share sufficient similarity to ensure sufficient knowledge transfer.

- *Decoupling Rationale*: While it is easy to understand how getting more data from an online simulator in the source domain is beneficial, the rationale behind the decision to maintain two separate world models (rather than a single, jointly trained model on both offline and online data) is not sufficiently explained or supported by evidence.

**Questions:**

- Have you considered training on multiple source domains?
- I would be curious to see experiments that use only one world model.
- Were all other approaches you compare to (except for the "Finetune" baseline) trained only on the offline data?

---

> ### Author Response · Authors · 2023-11-23
> **Responses to Reviewer G38A (Part 1)**
>
> Thank you for your effort in reviewing our paper. Below, you'll find our responses addressing your specific concerns.
>
> > Q1: The CoWorld framework's performance is critically dependent on the availability of an online simulator in a domain that is sufficiently similar to the target domain.
>
> Indeed, it is beneficial to leverage an online simulator as the source domain that is very similar to the target domain. However, it is essential to emphasize that (1) our approach does not assume a strict requirement for domain similarity, and (2) it is not that difficult to find a useful auxiliary simulator in practical applications.
>
> (1) Can CoWorld benefit from a less similar source domain?
>
> Our experimental results in **Figure 4** demonstrate CoWorld's ability to mitigate cross-environment discrepancies and benefit from a weakly related source domain. As presented in our **General Response (Q1)**, notable differences between the source (Meta-World) and target environments (RoboDesk) exist in visual observation ($\mathcal O$), physical dynamics ($\mathcal T$), reward definition ($\mathcal R$), and even the action space of robots ($\mathcal A$). CoWorld outperforms the second-best approaches by approximately 14%-18% in episode returns. In methodology, this is achieved by 1) Latent Space Alignment and 2) Target-Inclined Source Model Tuning.
>
> Furthermore, in **Figure 3(a)** in the revised paper, we have updated the Transfer Matrix among the 6 tasks on the Meta-World benchmark. There exist pairs of seemingly unrelated source and target tasks. Despite the challenging domain transfer setups, we observe that our approach outperforms Offline-DV2 in the majority of scenarios (26/30 cases).
>
> (2) How to choose the auxiliary online simulator given the offline target domain?
>
> Our results demonstrate the ability of CoWorld to mitigate the cross-domain discrepancies and leverage a less related online simulator to improve offline RL. This enhances the convenience of selecting an auxiliary simulator based on the type of robot (major) and task similarity (minor). For example, the Meta-World/RoboDesk/RLBench simulators can all be directly applied to the classic offline RL dataset, D4RL FrankaKitchen. Please refer to our **General Response (Q1)** for further details.
>
> > Q2: The rationale behind the decision to maintain two separate world models is not sufficiently explained or supported by evidence.
>
> (1) Rationale of separate world models:
>
> In light of non-identical observation spaces, diverse physics, varying reward scales, and even disparate action spaces between source and target MDPs (e.g., Meta-World and RoboDesk environments using different robotic arms with distinct action dimensions), we train separate world models for different domains. Our approach involves aligning the learned world models in latent state space and reward prediction. This strategy captures both domain-specific and shared information in the online-to-offline setup.
>
> (2) Empirical evidence for using separate world models vs. One world model:
>
> To investigate the effectiveness of employing separate world models, we conduct an experimental comparison between CoWorld and "Multi-Task DV2". The latter involves training DreamerV2 on both offline and online data with a joint world model and separate actor-critic models. Here, we present the mean returns and standard deviations for the cross-environment experiments (Meta-World to RoboDesk) that are calculated over 10 episodes with 3 random seeds. CoWorld consistently performs better.
>
> | Meta-World $\rightarrow$ RoboDesk               | Button Press  $\rightarrow$ Push Button | Window Close $\rightarrow$ Open Silde |
> |:------------------- |:-------------------------------------------- |:------------------------------------- |
> | Multi-Task DV2 (One world model) | 342 $\pm$ 29                                   | 173 $\pm$   22                        |
> | CoWorld             | 428 $\pm$ 42                                 | 202 $\pm$ 19                          |

---

> > ### Author Response · Authors · 2023-11-23
> > **Responses to Reviewer G38A (Part 2)**
> >
> > >Q3: Have you considered training on multiple source domains?
> >
> > Good question! CoWorld can be easily extended to the multi-source domain transfer setup. In the above **General Response (Q2)**, we have provided specific technical details of an adaptive source domain selection method, and the in-depth empirical analyses as well.
> >
> > We have included the multi-source experiments in the revision. Please see the following sections for details:
> > - We introduce the multi-source experimental setup in **Section 4.1**.
> > - We add new experimental results with multiple source domains to **Table 1**, **Figure 3**, and **Figure 4**, and provide more discussions in the main text.
> > - We provide the implementation details of adaptive source domain selection in **Appendix E**.
> >
> > >Q4: I would be curious to see experiments that use only one world model.
> >
> > Please refer to our response to Q2.
> >
> > >Q5: Were all other approaches you compare to (except for the "Finetune" baseline) trained only on the offline data?
> >
> > Yes, they were. However, to further demonstrate the effectiveness of CoWorld, we conducted additional experiments to compare CoWorld with new baselines that also involve the source data for model pretraining. Please refer to the results in our **General Response (Q3)**. From the results, we may conclude that a naive transfer learning method without any domain adaptation designs may introduce unexpected bias from the source domain and degenerate the target performance.

---

### Author Response · Authors · 2023-11-23
**General Responses to All Reviewers (Part 1)**

In addition to the specific responses below, we here reply to the general questions raised by the reviewers.

> Q1: Assumptions of the similarity between the source and target domains.

We acknowledge the reviewers' concern regarding the challenge of finding an online simulator sufficiently akin to the target domain. However, this assumption can be softened by our proposed approaches, namely, 1) latent space alignment and 2) target-inclined source model tuning. Through these approaches, we aim to mitigate domain discrepancies between distinct source and target MDPs.

Empirical evidence supporting our methods is presented in Section 4.3, where our proposed approach demonstrates robust performance in the "Meta-World to RoboDesk" transfer RL setup. We showcase the similarities and discrepancies of these two environments are presented below.

|              | Source: Meta-World                                   | Target: RoboDesk             | Similarity / Difference                                                                     |
|:------------ |:---------------------------------------------------- |:---------------------------- |:---------------------------------------------------------------------------------- |
| Task         | Window Close                                        | Open Slide                   | Related manipulation tasks |
| Dynamics     | Simulated Sawyer robot arm      | Simulated Franka Emika Panda robot arm      | Different                                                                   |
| Action space | Box([-1. -1. -1. -1.], [1. 1. 1. 1.], (4,), float64) | Box(-1, 1, (5,), np.float32) | Different                                                                   |
| Reward scale | [0, 1]                                               | [0, 10]                      | Different                                                                   |
| Observation  | Right-view images                               | Top-view images         | Different viewpoints, scene object, and backgrounds |

Our experiments reveal that the CoWorld method can effectively mitigate the cross-domain differences in visual observation, physical dynamics, reward definition, or even the action space of the robots. This characteristic makes it more convenient to choose an auxiliary simulator based on the type of robot. For example:
- When the target domain involves a robotic arm (e.g., RoboDesk), an existing robotic arm simulation environment (e.g., Meta-World as used in our paper) can be leveraged as the source domain.
- In scenarios with legged robots, environments like DeepMind Control with Humanoid tasks can serve as suitable auxiliary simulators.
- For target domains related to autonomous driving vehicles, simulation environments like CARLA can be selected.

It must be acknowledged that we have not yet conducted a theoretical analysis of the maximum tolerance for domain discrepancies. Nevertheless, we believe that the proposed method and experimental results presented in this paper remain practically significant.

---

> ### Author Response · Authors · 2023-11-23
> **General Responses to All Reviewers (Part 2)**
>
> > Q2: (1) Can CoWorld benefit from less related source tasks? (2) Can CoWorld deal with multi-source scenarios?
>
> (1) Clarification of CoWorld's performance with various source domains:
>
> We have updated the Transfer Matrix among 6 different tasks in **Figure 3(a)**, which presents the ratios of returns achieved by CoWorld compared to those achieved by the Offline DV2 without source domain pretraining. Values larger than 1 indicate positive domain transfers. We can see that our approach improves the offline RL performance in the majority of scenarios (26/30 cases).
>
> (2) Technical details and experimental analyses of Multi-Source CoWorld:
>
> CoWorld can easily accommodate multiple source domains by adaptively selecting suitable tasks as the auxiliary simulator. This extension can be easily achieved by measuring the latent space distance.
>
> **Method:** The key idea of adaptive domain selection is to allocate a set of one-hot weights $\omega_t^{i=1:N}$ to candidate source domains by calculating their KL divergence in the latent state space to the target domain, where $i \in [1,N]$ is the index of each source domain. The adaptive domain selection procedure includes the following steps:
> - **World Models Pretraining:** We pretrain a world model for each source domain individually.
> - **Domain Distance Measurement:** At each training step in the target domain, we measure the KL divergence between the latent states of the target domain, produced by $e^{(T)}\_{\phi}(o_t^{(T)})$, and corresponding states in each source domain, produced by $e^{(S)}_{\phi^{\prime}_i}(o_t^{(T)})$. Here, $e^{(T)}\_{\phi}$ is the encoder of the target world model and $e^{(S)}\_{\phi^{\prime}_i}$ is the encoder of the world model for source domain $i$.
> - **Auxiliary Domain Identification:** We dynamically identify the closest source domain with the smallest KL divergence. We set $\omega\_t^{i=1:N}$ as a one-hot vector, where $\omega\_t^i=1$ indicates the selected auxiliary domain.
> - **Rest of Training:** With the one-hot weights, we continue the rest of the proposed online-to-offline training approach. For example, in the representation learning stage, we adaptively align the target state space to the selected online simulator by rewriting the domain alignment loss term in Eq. (3) as
>
> $$
> \mathcal{L}\_{\text{M-S}}=\beta_2 \sum_{i=1}^N \omega_i \mathrm{KL}\left[\texttt{sg}(g(e_{\phi^{\prime}_i}^{(S)}\left(o\_t\right))) \ \| \ g(e\_{\phi}^{(T)}\left(o\_t\right)\right].
> $$
>
> **Results:** Below, we show the multi-source results on the cross-environment benchmark. We take three source domain tasks from Meta-World for adaptive source domain selection, while the target domain is from the RoboDesk environment.
>
> | Meta-World $\rightarrow$ RoboDesk | Source task(s) | Target: Push Button | Source task(s) | Target: Open Silde |
> |:--------------------------------- |:-------------- |:------------------- | -------------- | ------------------ |
> | Multi-Source CoWorld          | Button Press & Window Close             & Button Topdown   | 416 $\pm$ 33 | Window Close  & Drawer Close             & Handle Press | 204 $\pm$ 34        |     |    |
> | CoWorld w/ Manually Designated Source        | Button Press   | 428 $\pm$ 42        | Window Close   | 202 $\pm$ 19       |
> | Finetune                          | Button Press   | 359 $\pm$ 17        | Window Close   | 179 $\pm$ 8        |
>
> Additionally, we evaluate the multi-source CoWorld across multiple tasks within the Meta-World benchmark. Here, we use "Best-Source" to present the best results obtained on the target dataset among the 6 single-source CoWorld models using each individual source task as the auxiliary domain.
>
> | Target         | Multi-Source   | Best-Source    | Finetune        | Offline DV2    | Match Best-Source |
> | -------------- | -------------- | -------------- | --------------- | -------------- | :-----------------: |
> | Door Close     | 3546 $\pm$ 634 | 3967 $\pm$ 312 | 3500 $\pm$ 414  | 2143 $\pm$ 579 | &cross;           |
> | Button Press   | 3677 $\pm$ 476 | 3623 $\pm$ 543 | 3564 $\pm$ 417  | 3142 $\pm$ 533 | &check;           |
> | Handle Press   | 4460 $\pm$ 783 | 4570 $\pm$ 677 | 3702 $\pm$ 451  | 278 $\pm$ 128  | &check;           |
> | Button Topdown | 3626 $\pm$ 275 | 3921 $\pm$ 212 | 3499 $\pm$ 713  | 3002 $\pm$ 346 | &check;           |
> | Drawer Close   | 4841 $\pm$ 15  | 4845 $\pm$ 7   | 4273 $\pm$ 1327 | 3899 $\pm$ 679 | &check;           |
> | Window Close   | 4507 $\pm$ 59  | 4521 $\pm$ 367 | 4148 $\pm$ 971  | 3921 $\pm$ 752 | &cross;           |
>
> **Findings:** We observe that CoWorld can flexibly adapt to the transfer learning scenarios with multiple source domains, achieving comparable results to the model that exclusively uses our manually designated auxiliary simulator as the source domain (Best-Source). This study significantly improves the applicability of CoWorld in broader scenarios.
>
> We have included the above results in Table 1, Figure 3, and Figure 4 in the revised paper.

---

> > ### Author Response · Authors · 2023-11-23
> > **General Responses to All Reviewers (Part 3)**
> >
> > > Q3: More baselines trained with both the source domain and the target domain.
> >
> > As suggested by the reviewers, we have compared CoWorld with more baseline models that include the source data in the training process. Specifically, for all of the baseline models below, we pretrained the models using source domain data for 160k steps, and finetuned them in the target domain for another 300k steps. These experiments are conducted on the Meta-World benchmark, and so the following results can be seen as a supplement to the results in Table 1.
> >
> > | Model                | BP $\rightarrow$ DC* | DC $\rightarrow$ BP | BP $\rightarrow$ HP | HP $\rightarrow$ BT |  DC* $\rightarrow$ DC | DC $\rightarrow$ WC | Avg. |
> > |----------------------|:--------------------------------:|:-------------------:|:-------------------:|:-------------------:|:------------------------------:|:-------------------:|:----:|
> > | Offline DV2 w/ source | 189 $\pm$ 100    |  1089 $\pm$ 528   |   765 $\pm$ 400   | 1052 $\pm$ 544  |    3759 $\pm$ 2457   |   1717 $\pm$ 958  |  1429    |
> > | DrQ+BC w/ source      |       38 $\pm$ 113                           |       22 $\pm$ 50             |   254 $\pm$ 217 |  95 $\pm$ 114  | 419 $\pm$ 134    |   233 $\pm$ 192   |   177   |
> > | LOMPO w/ source     |  236 $\pm$ 23     |  169 $\pm$ 60   |    16 $\pm$ 13   |  478 $\pm$ 87     |   1415 $\pm$ 2156    |   195 $\pm$ 79 |  418  |
> > | Finetune+EWC |      2479 $\pm$ 1006   |      47 $\pm$ 10               |       314 $\pm$ 393             |        232  $\pm$ 118           |     825 $\pm$ 1665                           |         451 $\pm$ 55          |   736   |
> > | CoWorld     | 3967 $\pm$ 312                             | 3623 $\pm$ 543               | 4570  $\pm$ 677             |3921 $\pm$ 212| 4515 $\pm$ 24  | 4521    $\pm$ 367                       |  4241 |
> >
> > **Findings:** As we can see from the above results, all of the baseline models suffer from the so-called "negative transfer" effect when incorporating a source pretraining stage. For instance, the LOMPO model achieves an average return of 1712 when it is trained from scratch in the target domain, while it achieves an average return of 418 when it involves the source data for model pretraining. It is an interesting observation, as it implies that **a naive transfer learning method may degenerate the target performance by introducing unexpected bias**.
> >
> > **Finetune+EWC:** Particularly, we implemented a baseline model that incorporates EWC, a classic continual learning method, to regularize the model for retaining knowledge from the online source domain. However, we observed that without additional model designs for domain adaptation, retaining the source domain knowledge may ultimately impact the performance in the target domain. We have included the above results in Table 1 in the revised paper.
> >
> > > Q4: Time complexity.
> >
> > We evaluate the training/inference time on the Meta-World benchmark (Handle Press $\rightarrow$ Button Topdown) using an RTX 3090 GPU:
> > | Model |Number of Training Iterations| Training Time | Inference Time Per Episode |
> > |:--------------------------------- |:-------------: | :---:| :---:|
> > | OfflineDV2                        | 300k |2054 min     |                     2.95 sec |
> > | DrQ+BC                            | 300k|200 min      |                     2.28 sec |
> > | CQL                               | 300k|405 min      |                     1.88 sec |
> > | CURL                              |300k| 434 min      |                     2.99 sec |
> > | LOMPO                             | 100k|1626 min     |                     4.98 sec |
> > | Finetune                          | 460k|1933 min |                     6.63 sec |
> > | Finetune+EWC                      |  460k|    1533 min         |                     5.58 sec |
> > | CoWorld                           | 460k|3346 min  |                     4.47 sec |

---

### Author Response · Authors · 2023-11-23
**Revision Uploaded**

We thank all reviewers for the constructive comments and have updated our paper accordingly. Please take a moment to check out the revised manuscript, which incorporates the following key modifications:

**New Results:**
1. Evaluated CoWorld in the multi-source experimental setup (Table 1 and Figure 4).
2. Added a new baseline model, "Finetune+EWC", with a classic continual learning method (Table 1).
3. Refined the transfer matrix across various pairs of source and target domains, showing the effect of using less related source tasks (Figure 3).
4. Added the value estimation results of DrQ-v2, which is the base model of CQL (Figure 5).
5. Extended the hyperparameter analysis about $k$ with finer intervals (Figure 11).

**Other Revisions:**
1. Refined the introduction to elaborate on the paper’s contributions (Section 1).
2. Modified the adaptive source domain selection method to extend CoWorld to scenarios with multiple source domains available (Section 4.1 and Appendix E).
3. Included more details of the architecture and the objective functions of our approach (Appendix A.2 and A.3).
4. Added implementation details of the Finetune and Finetune+EWC baselines (Appendix B).
5. Clarified our assumptions for the similarity between the source and target domains (Appendix D).
6. Added the technical details of the adaptive source domain selection method (Appendix E).
7. Corrected the identified typos.


Please do not hesitate to let us know for any additional comments on the paper.

---

### Meta-Review · Area_Chair_uwoU · 2023-12-12

**Metareview:**

### Summary
This paper introduces a novel approach to address challenges in training offline reinforcement learning (RL) models using visual inputs. It tries to tackle the issues of overfitting and overestimation by leveraging an RL simulator as an auxiliary domain for online interaction. By aligning the value model to target data, the online simulator serves as a "test bed" for offline policies, allowing for conservative value estimation without hindering exploration. The authors claim their proposed method significantly outperforms existing offline visual RL approaches in challenging environments, as demonstrated through experiments in DeepMind control suite, Meta-World, and RoboDesk environments.

### Decision
Unfortunately, this paper is not well-written, although the idea presented is interesting. As a result, this resulted in the high variance scores between the reviewers. Currently, this paper is not ready for publication at ICLR. I would recommend the authors revise the paper, taking into consideration the feedback provided by the reviewers, and consider resubmitting to another venue in the future. Let me summarize the weaknesses highlighted by the reviewers:

1. Dependence on Auxiliary Simulators

Reviewer G38A: The CoWorld framework heavily relies on online simulators, limiting applicability when such simulators aren't available or domains lack similarity.
Reviewer mTFC: The success of the online-to-offline paradigm hinges on source-target domain similarity, limiting generalizability in real-world scenarios lacking similar domains

2. Lack of Explanation for Decoupling Rationale

Reviewer G38A: The decision to maintain separate world models isn't adequately explained or supported.
Reviewer mTFC: The rationale behind maintaining two separate world models instead of a single jointly trained model lacks clarity and supporting evidence.

3. Marginal or Unclear Performance Improvement

Reviewer mTFC: CoWorld's performance improvement is questioned due to sensitivity to hyperparameters, computational complexity, and marginal gains compared to alternative methods.
Reviewer MD4J: The paper lacks clarity in distinguishing CoWorld as a method versus a new problem statement, leading to insufficient baselines and evaluation against different RL algorithms.
4. Lack of Clarity in Source Domain Selection and Relation

Reviewer wybs: The practicality of source domain selection is questioned, as it relies on information unavailable during training, potentially leaking test-time information into the process.
Reviewer wybs: Insufficient clarity regarding the relationship between source and target domains, impacting the method's feasibility.

5. Computational Complexity and Stability
Reviewer mTFC: The alternation between online and offline agents until convergence is highlighted as potentially expensive and unstable, lacking evidence of time complexity compared to baselines.

These weaknesses undermine the paper's contribution due to unrealistic assumptions, methodological limitations, lack of clarity in rationale and implementation, and inadequate evaluation against alternative approaches and baselines. Therefore, the paper is being rejected based on these substantial issues raised by the reviewers (G38A, mTFC, MD4J, wybs).

**Justification For Why Not Higher Score:**

Two reviewers recommended this paper for clear rejection. In its current form, this paper is not ready for publication yet.

**Justification For Why Not Lower Score:**

N/A

---

### Decision · Program_Chairs · 2024-01-16

Reject